# Mechanism-based rescue of Munc18-1 dysfunction in varied encephalopathies by chemical chaperones

Noah Guy Lewis Guiberson[1], André Pineda[1], Debra Abramov[1], Parinati Kharel[1], Kathryn E. Carnazza[1], Rachel T. Wragg[2], Jeremy S. Dittman[2] & Jacqueline Burré[1]

Heterozygous de novo mutations in the neuronal protein Munc18-1 are linked to epilepsies, intellectual disability, movement disorders, and neurodegeneration. These devastating diseases have a poor prognosis and no known cure, due to lack of understanding of the underlying disease mechanism. To determine how mutations in Munc18-1 cause disease, we use newly generated *S. cerevisiae* strains, *C. elegans* models, and conditional Munc18-1 knockout mouse neurons expressing wild-type or mutant Munc18-1, as well as in vitro studies. We find that at least five disease-linked missense mutations of Munc18-1 result in destabilization and aggregation of the mutant protein. Aggregates of mutant Munc18-1 incorporate wild-type Munc18-1, depleting functional Munc18-1 levels beyond hemizygous levels. We demonstrate that the three chemical chaperones 4-phenylbutyrate, sorbitol, and trehalose reverse the deficits caused by mutations in Munc18-1 in vitro and in vivo in multiple models, offering a novel strategy for the treatment of varied encephalopathies.

[1] Brain and Mind Research Institute & Appel Institute for Alzheimer's Disease Research, Weill Cornell Medicine, New York, NY, 10021, USA. [2] Department of Biochemistry, Weill Cornell Medicine, New York, NY 10021, USA. These authors contributed equally: Noah Guy Lewis Guiberson, André Pineda. Correspondence and requests for materials should be addressed to J.Bé. (email: jab2058@med.cornell.edu)

Heterozygous de novo mutations in the neuronal protein Munc18-1 (also called STXBP1) were first described in 2008 to cause the infantile epileptic encephalopathy Ohtahara syndrome[1]. Since then, mutations in Munc18-1 have been linked to a spectrum of neuronal disorders, including West syndrome[2], Dravet syndrome[3], Lennox–Gastaut syndrome[4], non-syndromic epilepsy, focal seizures with neonatal onset[5], Rett syndrome[6], and intellectual disability without epilepsy[7]. Furthermore, a variety of associated movement disorders like ataxia, tremor, head tremor, and juvenile-onset parkinsonism were described in patients with Munc18-1 mutations[8–11], and dysregulation of Munc18-1 expression levels are associated with Alzheimer's disease[12,13]. Medical management of seizures and developmental impairments is difficult since these diseases are largely refractory to standard anti-epileptic drugs (reviewed in ref. [14]). Similarly, ataxia, tremor, and neurodegeneration in patients with Munc18-1 mutations are intractable to treatment. So far, no therapy has shown significant long-term improvements, and severe morbidity and high mortality are the inevitable outcomes in some of these diseases.

SEC1/Munc18-like proteins are essential for secretion in yeast (SEC1[15]), *Caenorhabditis elegans* (UNC-18[16]), zebrafish (Stxbp1[17]), *Drosophila melanogaster* (rop[18]), and in mice (Munc18-1[19]). In yeast, SEC1 mutations block secretion, resulting in accumulation of secretory vesicles[20]. In *C. elegans*, although viable, *unc-18* null animals are paralyzed, and exhibit a reduced primed vesicle pool and severe defects in locomotion and neurotransmitter release[16,21,22], while heterozygous *unc-18* worms reveal no impairments in neurotransmitter release[23]. In zebrafish, knockout of *stxbp1a* or *stxbp1b* causes seizures and defects in development, locomotor activity, and metabolic rate[17]. Rop null mutants exhibit morphological defects and die as embryos[18], while heterozygous rop mutants are viable and display decreased synaptic activity[24]. In mice, knockout of Munc18-1 is lethal, and abolishes neurotransmitter release in cultured neurons[19]. Heterozygous mice are viable and display normal synaptic vesicle fusion, but reveal a reduction in the readily releasable pool of synaptic vesicles[25]. Together, these data define a critical regulatory function of Munc18-1 in neurotransmitter release, in particular in determining the number of readily releasable vesicles, and raise the possibility that Munc18-1 mutations in humans cause severe disease not only by a loss-of-function mechanism, i.e., haploinsufficiency, but by asserting an additional dominant-negative effect on the wild-type allele.

It is widely assumed that Munc18-1-linked disorders are caused by haploinsufficiency, due to the occurrence of heterozygous missense mutations, nonsense mutations, frame shifts, and deletions[10]. Yet, recently, a dominant-negative effect was proposed, based on overexpression of a GFP-tagged variant of Munc18-1[26]. Heterozygous mice, flies, and worms show no epileptic or developmental phenotype[23–25]. At the same time, heterozygous neurons generated from human embryonic stem cells display a reduction in excitatory post-synaptic currents[27]. Recent studies have suggested that mutations in Munc18-1 could result in a thermo-labile protein[28], and temperature-sensitive structural changes associated with the C180Y mutation have been reported for a GFP-tagged C180Y variant in PC12 cells[29]. Thus, it remains unclear how mutations in Munc18-1 cause varied autosomal-dominant disorders, and a systematic and detailed understanding of their etiology is needed in order to develop effective strategies to counteract their deleterious effects.

Here, we find that missense mutations of Munc18-1 result in destabilization and aggregation of the mutant protein. We use newly generated *S. cerevisiae* strains, *C. elegans* models, conditional Munc18-1 knockout mouse neurons expressing wild-type or mutant Munc18-1, as well as in vitro studies, and demonstrate that mutant Munc18-1 recruits endogenous wild-type Munc18-1 into insoluble aggregates, depleting functional Munc18-1 levels beyond hemizygous levels. Importantly, we demonstrate that the three chemical chaperones 4-phenylbutyrate, sorbitol, and trehalose are able to stabilize Munc18-1 protein levels, reversing the insolubility and aggregation of mutant Munc18-1, and to rescue neuronal deficits in vitro and in vivo, providing a novel therapeutic approach for Munc18-1-associated encephalopathies.

## Results

**Disease-linked mutations in Munc18-1.** When we analyzed the distribution of disease-linked missense mutations in Munc18-1 in its primary and secondary sequence, we found no specific area or domain of Munc18-1 to be particularly affected (Supplementary Fig. 1), suggesting loss of function of Munc18-1 mutants as the underlying disease mechanism. To test whether haploinsufficiency is the cause of disease, we studied the impact of the five disease-linked missense mutations P335L, R406H, P480L, G544D, and G544V on Munc18-1 structure and function (highlighted in blue in Supplementary Fig. 1). We chose these five mutants because: (1) multiple mutations at these residues are associated with disease (Supplementary Fig. S1 and ref. [10]), (2) disease-linked residues are conserved in the *C. elegans* homolog UNC-18 (Supplementary Fig. 2a), permitting the study of motor phenotypes in our newly generated *C. elegans* disease models in vivo, (3) homologous residues are mutated in Munc18-2 in the immune disease familial hemophagocytic lymphohistiocytosis type 5[30], enabling extension of our findings to another organ system and disease, (4) residue P335 has been proposed to function as a flexible hinge point in Munc18-1, modulating the binding of Munc18-1 to syntaxin-1, VAMP2, and the SNARE complex, and regulating neurotransmitter release[31–33], and (5) we maintained expression of full-length protein while replicating disease-linked mutations to enable antibody-based detection.

**Neuronal impairments in *C. elegans* expressing *unc-18* mutants.** Epileptic syndromes are linked with synaptic firing abnormalities, likely due to an inappropriate inhibitory tone[34,35], and synaptic abnormalities are common in Rett syndrome, intellectual disabilities, movement disorders, and neurodegeneration[6–12]. Munc18-1 plays a critical regulatory function in neurotransmitter release, in particular in determining the number of readily releasable vesicles (reviewed in ref. [36]), and mutations in Munc18-1 can be expected to affect this central function, leading to the observed phenotypes in affected patients.

To assess the effect of mutations in Munc18-1 on Munc18-1 function in the nervous system in vivo, we introduced analogous mutations in the *C. elegans unc-18* sequence (Supplementary Fig. 2b), and generated transgenic nematodes expressing *unc-18* wild-type (WT), R405H, P479L, and G544D in an *unc-18* null background. Neuronal function was assessed using two behavioral assays: worm locomotion[37] and the aldicarb sensitivity assay[38]. We first confirmed complete rescue of the paralysis of *unc-18* null worms[16] by re-introducing WT *unc-18* (Fig. 1a). In contrast, all three mutants displayed a >30% reduction in locomotion compared to WT *unc-18* (Fig. 1b). The mutant proteins were expressed at equal or higher levels compared to WT, indicating that the reduced rescue was not due to lower expression of mutant UNC-18 (Supplementary Fig. 2c).

Defective locomotion is only an indirect qualifier for synaptic dysfunction. We thus measured neurotransmitter release at the neuromuscular junction in these worms using a pharmacological approach. *C. elegans* use the presynaptic release of the neurotransmitters GABA and acetylcholine to stimulate muscle contraction and drive locomotion. Exposing worms to

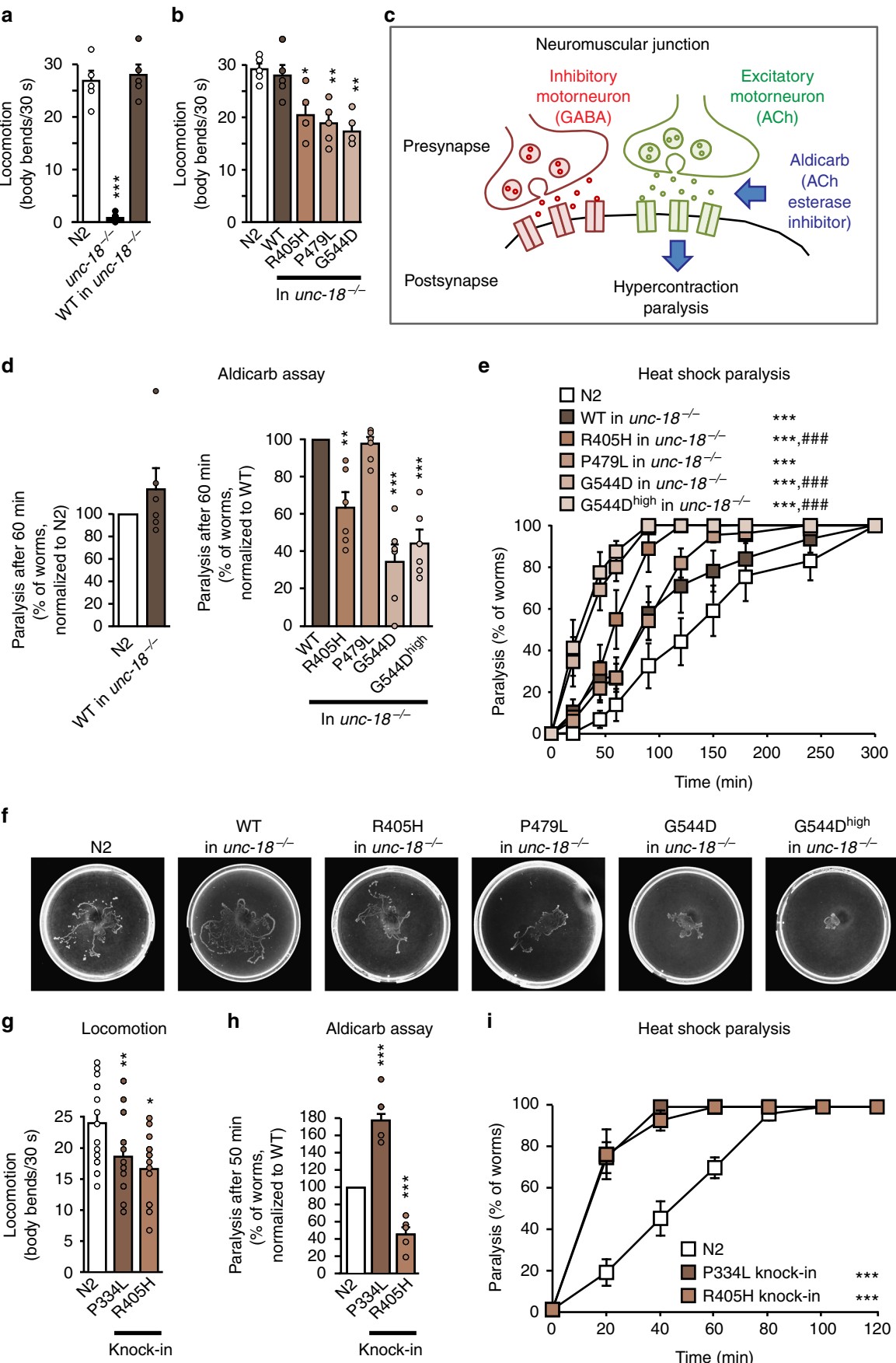

**Fig. 1** Neuronal impairments in *C. elegans* expressing mutant UNC-18. **a**, **b** Locomotion of *C. elegans*. Body bends of indicated worm strains per 30 s were counted. Data are means ± SEM (*$p < 0.05$, **$p < 0.01$, ***$p < 0.001$ by Student's *t* test; $n = 5$ independent experiments on ten worms per experiment). **c** Scheme of the aldicarb assay. **d** Mutants display reduced acetylcholine release at the worm neuromuscular junction. Paralysis of young adult worms expressing WT or mutant *unc-18* was measured 60 min after exposure to aldicarb (see Supplementary Fig. 2d for entire curves). Data are means ± SEM (**$p < 0.01$, ***$p < 0.001$ by Student's *t* test; $n = 6$ independent experiments on 20 worms per experiment). **e** Heat-induced paralysis. Indicated worm strains were exposed to 37 °C over a period of 300 min, and paralysis was scored at indicated time points. Data are means ± SEM (***$p < 0.001$ by two-way ANOVA, compared to N2; ###$p < 0.001$ by two-way ANOVA, compared to *unc-18*-WT in *unc-18*$^{-/-}$, $n = 9$–13 independent experiments on ten worms per experiment). **f** Worm traces after heat-induced paralysis. Plates were imaged after heat shock analysis in **e**. **g**–**i** Locomotion, acetylcholine release, and heat shock paralysis of CRISPR-edited *C. elegans*. Worms were assayed as in **a**–**e**. Data are means ± SEM (*$p < 0.05$, **$p < 0.01$, ***$p < 0.001$ by Student's *t* test in **g** and **h** and two-way ANOVA in **i**; $n = 15$–20 worms for **g**, $n = 5$–6 independent experiments on 20–25 worms per experiment for **h**, and $n = 6$ independent experiments on ten worms per experiment for **i**)

acetylcholine esterase inhibitors such as aldicarb lead to the inability to turn off acetylcholine signaling at the neuromuscular junction, causing hypercontraction and eventual paralysis (Fig. 1c). While *unc-18* null worms rescued with WT *unc-18* revealed no significant difference compared to N2 worms, worm strains expressing R405H and G544D *unc-18* exhibited a severe impairment in neurotransmitter release, and paralyzed more slowly (Fig. 1d and Supplementary Fig. 2d). We did not detect a significant impairment in the P479L worm strains, likely because the R405H and G544D mutations exhibit stronger phenotypes (compare Fig. 1d, e), and the aldicarb assay may not be sensitive enough to detect the synaptic effects of the P479L mutation.

To probe for UNC-18 misfolding and the suggested thermolability of mutant UNC-18[28], we analyzed paralysis of worms under heat shock. Worms expressing mutant *unc-18* variants paralyzed significantly faster compared to either N2 *C. elegans* or *unc-18* null worms rescued with WT *unc-18* (Fig. 1e), and exhibited reduced locomotion during heat shock (Fig. 1f).

One caveat of overexpression of UNC-18 variants is the potential of anti-, hyper-, or neomorphic gain-of-function effects, in addition to missing a functional deficit because enough functional mutant protein is expressed. This may also explain the mild phenotype of the P479L mutation in above experiments. We thus generated P334L and R405H knock-in worms using CRISPR/Cas9 technology (Supplementary Fig. 2e and 2f) and repeated the same analyses. Similar to our transgenic *C. elegans* models, we found a reduction in locomotion, neurotransmitter release, and an exaggerated paralysis in response to heat shock in R405H worms compared to N2 worms (Fig. 1g–i). The mutant phenotype of our knock-in worms was stronger compared to the transgenic *C. elegans*, suggesting that overexpression of mutant UNC-18 variants partially compensated for the loss of function. The P334L mutant revealed similar impairments in locomotion and heat shock paralysis, but had an increase in neurotransmitter release (Fig. 1g–i). Taken together, disease-linked variants of *unc-18* result in significant impairments in nervous system function in vivo.

**Synaptic impairments in primary mouse neurons.** Our in vivo studies in *C. elegans*, combined with the importance of UNC-18 in synaptic release (reviewed in ref. [36]), strongly suggest a synaptic impairment caused by mutations in UNC-18 that causes the varied encephalopathy phenotypes. To dissect the molecular mechanism underlying these deficits, we tested the effect of mutations in Munc18-1 in conditional Munc18-1 knockout mice, in which exon 2 of the *Munc18-1* gene is flanked by loxP sites and can be excised using cre recombinase[39]. We generated primary cortical neurons from these mice and infected them with lentiviral vectors expressing cre recombinase to drive knockout of Munc18-1, or with an inactive version of cre (Δcre) as control[40] (Supplementary Fig. 3a). We then re-introduced myc-tagged WT Munc18-1 via lentiviral expression, and found that the lentiviral

transduction resulted in moderate expression levels of Munc18-1, close to WT levels and not overexpression, even upon addition of exorbitant lentivirus amounts (Supplementary Fig. 3a and 3b).

We then assessed the ability of WT and mutant Munc18-1 to regulate neurotransmitter release in these neurons using a microelectrode array (Fig. 2a–g). We measured individual neuronal activity as well as network activity in primary Munc18-1 knockout neurons and neurons expressing WT or mutant Munc18-1 variants (Fig. 2b–d). Compared to WT Munc18-1, R406H, P480L, G544D, and G544V caused a reduction in the mean firing rate, network burst frequency, and network burst percentage, similar to levels of cre-infected neurons (Fig. 2c–g and Supplementary Fig. S3c–e). P335L revealed no significant phenotype in this assay, but showed a trend toward increased activity (Fig. 2d–g and Supplementary Fig. S3c–e), similar to our *C. elegans* data (Fig. 1h).

In parallel, we measured synaptic vesicle recycling using an antibody uptake assay[41]. We stimulated primary Munc18-1 null neurons expressing WT or mutant Munc18-1 with high potassium and quantified the fluorescence intensity of endocytosed synaptotagmin-1 antibody (Fig. 2h). As controls, we used primary neurons expressing cre alone in high potassium solution, or Δcre in low potassium solution. Both control conditions resulted in negligible uptake of antibody, indicated by minimal levels of synaptic vesicle cycling (Fig. 2h, i and Supplementary Fig. 3f). In contrast, Munc18-1 null neurons rescued with WT Munc18-1 revealed a pronounced uptake of antibody, while all of the disease-linked mutations showed a significantly reduced uptake relative to WT (Fig. 2h, i and Supplementary Fig. 3f). These data suggest that mutations in Munc18-1 impair neurotransmitter release and synaptic vesicle recycling, supporting our *C. elegans* data (Fig. 1).

Taken together, we found severe impairments in synapse function in both primary neurons and in vivo when expressing disease-causing mutations in Munc18-1/UNC-18. We next addressed the question of what the underlying molecular mechanisms of these functional deficits are.

**Mutants of Munc18-1 are metabolically unstable.** Mutations in Munc18-1 may impair its function by reducing protein levels. We thus first measured the total protein levels in primary Munc18-1 knockout neurons expressing WT or disease-linked mutants of Munc18-1. Compared to WT Munc18-1, all mutants exhibited a significant reduction in protein levels, with four out of the five mutants resulting in protein levels below 10% of WT levels (Fig. 3a). This change was not due to changes in mRNA levels (Supplementary Fig. 4a–c).

Reduced levels of a protein containing a point mutation, especially with the higher thermal sensitivity in our in vivo function studies (Fig. 1e, i), led us to hypothesize that mutant Munc18-1 is rapidly turned over. To measure Munc18-1 stability, we blocked protein translation using cycloheximide in primary

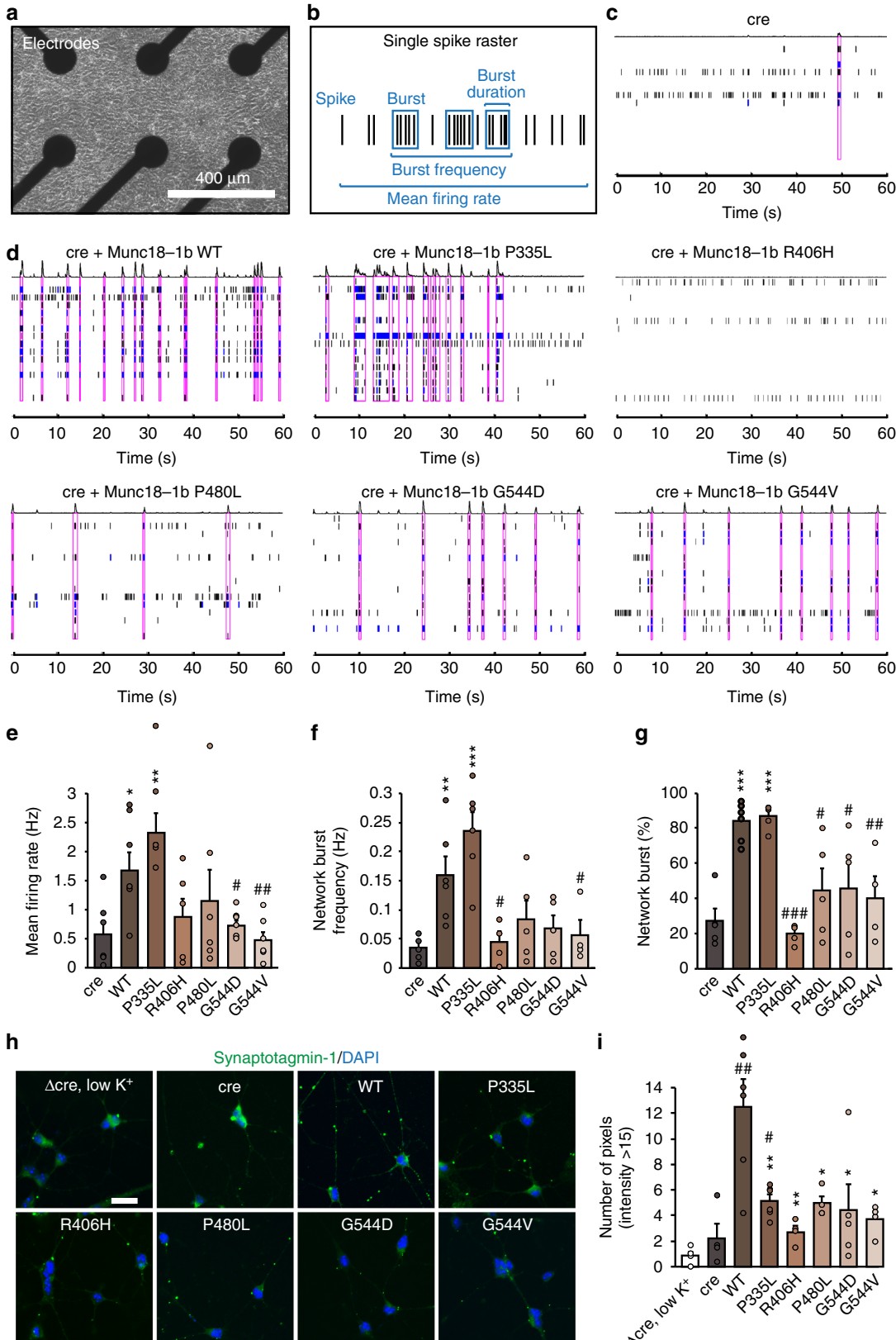

Munc18-1 knockout neurons expressing WT or mutant Munc18-1. We found WT Munc18-1 to be significantly more stable compared to mutant variants (Fig. 3b). Since detectable protein levels for the Munc18-1 mutants were very low in neurons, we repeated the same experiments in heterologous cell lines. We found identical reductions in mutant Munc18-1 levels in transfected HEK293T and Neuro2a neuroblastoma cells (Supplementary Figs. 4d and 4e), enabling the use of these cell lines for testing the effect of mutations on protein stability. We then repeated the cycloheximide chase experiment in HEK293T cells,

**Fig. 2** Synaptic impairments in neurons expressing mutant Munc18-1. **a** Primary neurons were plated on a multi-electrode array. **b–g** Munc18-1 knockout neurons (**c**) or knockout neurons expressing Munc18-1b variants (**d**) were subjected to analysis of mean firing rate, burst frequency, burst duration, and network burst activity (**b**, **e–g**). Purple boxes in **c** and **d** indicate network activity. Data are means ± SEM (*,#$p < 0.05$, **,##$p < 0.01$, ***,###$p < 0.001$ by Student's $t$ test; $n = 4–6$ independent experiments). **h**, **i** Uptake of synaptotagmin-1 antibody during high $K^+$ stimulation. Neurons expressing cre recombinase and/or WT or mutant Munc18-1b were subjected to an antibody uptake assay. Endocytosed synaptotagmin-1 antibody was quantified by immunostaining (**h**), via counting the number of pixels > intensity of 15 (**i**). Data are means ± SEM (*,#$p < 0.05$, **,##$p < 0.01$ by Student's $t$ test; $n = 4$ independent experiments). Scale bar in **h** = 20 μm

and found that all Munc18-1 mutants were turned over significantly faster compared to WT Munc18-1 (Supplementary Fig. 4f).

Although cycloheximide at the used concentration and application time is not toxic to neurons[42], cycloheximide non-specifically blocks translation of all proteins, including interactors of Munc18-1, which may affect its turnover. In parallel, we thus measured turnover of Munc18-1 variants using a Dendra2 photoconversion assay. We expressed Dendra2-tagged Munc18-1 fusion proteins in primary neurons and measured the rate of disappearance of the converted red signal after photoactivation (Supplementary Fig. 4g–j). Similar to the cycloheximide chase experiments, we found mutant Munc18-1 to turn over significantly faster compared to WT Munc18-1 (Supplementary Fig. 4i and 4j). Due to low mutant Munc18-1 protein levels and high signal/noise ratio in these imaging experiments, we repeated the same assay in transfected HEK293T cells which also allowed us live imaging of same cells over time due to overall higher signal strength (Fig. 3c). Similar to primary neurons and at similar photoconversion rates, mutant Munc18-1 variants turned over significantly faster compared to WT (Fig. 3c–f).

Overall, these data support a direct effect of mutations on the metabolic stability of Munc18-1, and suggest structural destabilization of Munc18-1 as an underlying cause of Munc18-1-linked encephalopathies.

**Mutations in Munc18-1 promote insolubility and aggregation.** Structural instability and misfolding of mutant Munc18-1 may not only cause rapid turnover of Munc18-1, but may also be accompanied by Munc18-1 aggregation, as already indicated by the photoconversion experiments (Fig. 3c and Supplementary Fig. 4g). To directly test whether mutations in Munc18-1 result in Munc18-1 aggregation, we first measured solubility of Munc18-1 variants biochemically using the non-ionic detergent Triton X-100. We found a significant reduction in the solubility for mutant Munc18-1 in primary neurons (Fig. 4a). Due to low expression of mutant Munc18-1, we confirmed these findings in transfected HEK293T cells and Neuro2a cells (Supplementary Figs. 5a and 5b).

Second, we measured susceptibility of Munc18-1 mutants to trypsin cleavage as a means of analyzing in situ protein aggregation in native conditions, in the absence of detergent. If mutant Munc18-1 forms aggregates, accessibility of tryptic cleavage sites may be reduced, rendering the protein less susceptible to tryptic digestion. In primary Munc18-1 knockout neurons rescued with WT or mutant Munc18-1 as well as in transfected Neuro2a and HEK293T cells, we found Munc18-1 mutants to be more resistant to tryptic digest, suggesting protein aggregation (Fig. 4b and Supplementary Figs. 5c and 5d).

Third, we assessed Munc18-1 mutants for the formation of visible aggregates, and found juxtanuclear and neuritic aggregates in primary cortical Munc18-1 knockout neurons when Munc18-1 mutants were expressed, but not WT Munc18-1 (Fig. 4c). Aggregates increased in size over time, starting with small foci localized throughout the cytoplasm that grew into big juxtanuclear aggregates (Supplementary Fig. 6a) and lacked co-localization with endoplasmic reticulum, Golgi complex,

lysosomes, or mitochondria in Neuro2a cells (Supplementary Figs. 6b–f). Although these insoluble aggregates were not cleared at the same rate as soluble Munc18-1 mutants, as assessed by a cycloheximide chase experiments of Triton X-100 soluble and insoluble fractions in Neuro2a cells (Supplementary Figs. 7a and 7b), aggregates were not toxic, as determined by an MTT assay (Supplementary Figs. 7c and 7d) and counting of neuronal nuclei in primary neurons and in Neuro2a cells (Supplementary Fig. 7e).

Does mutant Munc18-1 form similar aggregates in vivo? To test this, we generated *C. elegans* strains expressing GFP-tagged WT and G544D *unc-18* in an *unc-18* null background. To exclude a potential effect of the GFP-tag on neuron function, we first compared both strains to N2 worms. We found WT::GFP to fully rescue the *unc-18* paralysis and deficits in neurotransmitter release, while the G544D::GFP variant did not, despite similar expression levels (Fig. 4d, e and Supplementary Figs. 8a–c), similar to the untagged variants we had analyzed (Fig. 1b). We next assessed worms for the subcellular localization of WT::GFP and G544D::GFP in the worm ventral nerve cord. Typical for a soluble protein, we found WT::GFP to be distributed throughout the ventral nerve cord, whereas the G544D::GFP mutant resulted in a punctate pattern with loss of axonal localization (Fig. 4f). Interestingly, G544D::GFP accumulated either in pairs of two bigger puncta, or as single slightly smaller puncta (Fig. 4f), reminiscent of the mutant Munc18-1 aggregates in primary neurons and in Neuro2a cells (Fig. 4c and Supplementary Fig. 6). This phenotype was accompanied by reduced locomotion and an accelerated paralysis in the aldicarb and heat shock assays (Fig. 4d, e, g and Supplementary Figs. 8d and 8e), suggesting loss of function due to rapid protein degradation and/or aggregation.

In parallel, we generated *S. cerevisiae* strains expressing GFP-tagged or untagged variants of Munc18-1, because of the highly conserved function of Munc18-1/SEC-1 across various model species and the ease to manipulate and screen yeast. Strikingly, we found the same aggregation phenotype and the same reduction in mutant protein levels in yeast when compared to heterologous cell lines, primary neurons and *C. elegans* (Fig. 4h–j and Supplementary Fig. 9), and confirmed the lack of toxicity of mutant Munc18-1 also in yeast (Supplementary Fig. 9).

Overall, we found that mutations in Munc18-1 result in decreased protein levels due to rapid protein degradation, and in significant aggregation of remaining protein that is resistant to clearance, suggesting severe misfolding of Munc18-1 mutants. This fate of mutant Munc18-1 points to loss of function and haploinsufficiency as the possible cause of disease. However, lack of a major synaptic and developmental phenotype in heterozygous mice, flies, and worms[23–25] suggests that haploinsufficiency may be insufficient as explanation, and the severe autosomal-dominant encephalopathies may be a result of a loss of >50% of Munc18-1 function. We thus explored the possibility of a dominant-negative disease mechanism, where mutant Munc18-1 negatively affects the WT Munc18-1 protein.

**Mutations in Munc18-1 cause a dominant-negative phenotype.** To test a dominant-negative disease mechanism, we expressed mutant Munc18-1 in the presence of WT Munc18-1. We first

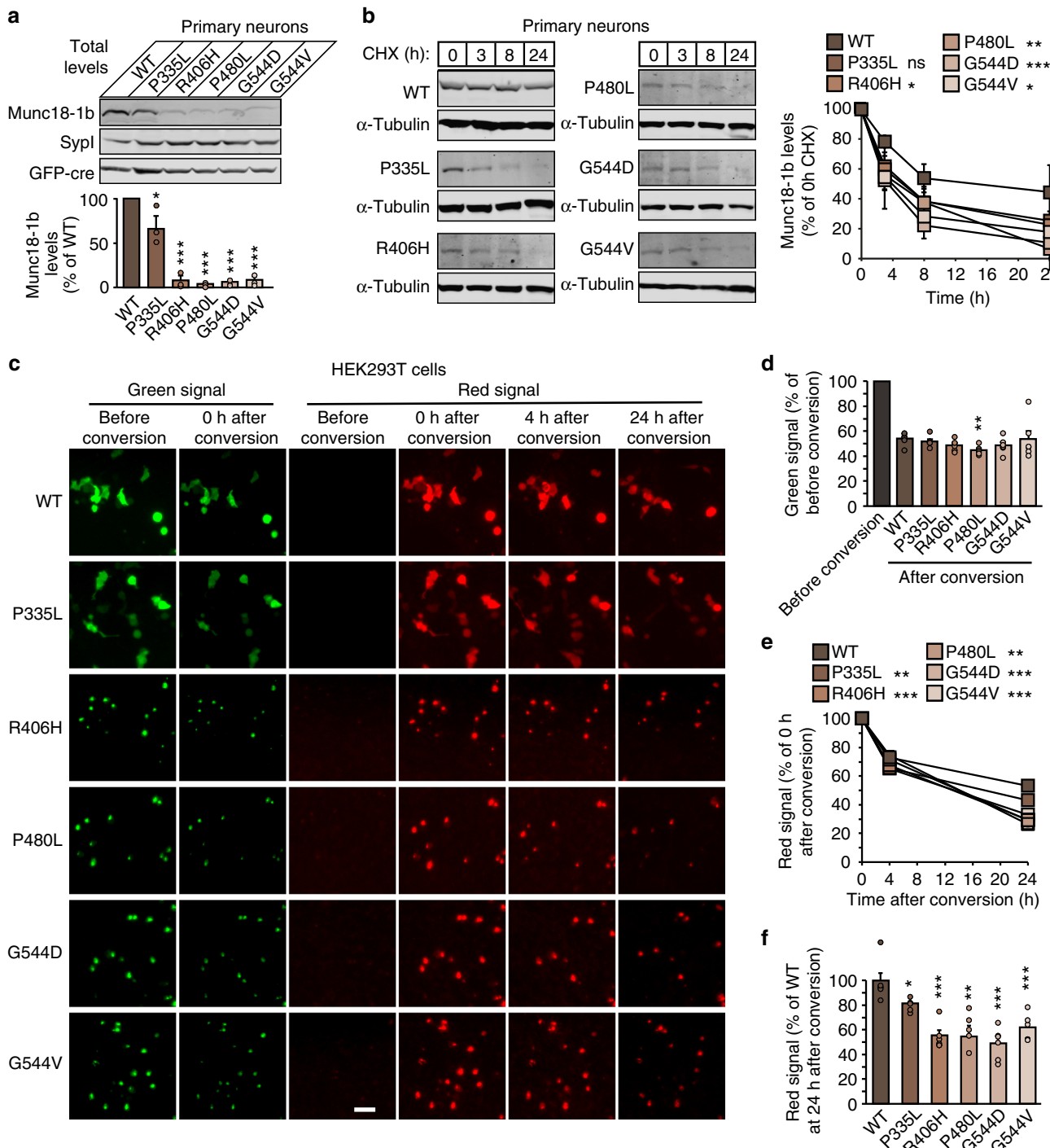

**Fig. 3** Increased turnover of Munc18-1 mutants in neurons. **a** Total protein levels of Munc18-1. WT and mutant Munc18-1b were expressed in primary neurons infected with lentiviral vectors expressing cre recombinase. Total protein levels were quantified by immunoblotting, normalized to the levels of the synaptic protein synaptophysin-1 (SypI). Data are means ± SEM (*$p < 0.05$, ***$p < 0.001$ by Student's $t$ test; $n = 3$ independent experiments). **b** Turnover of Munc18-1 by cycloheximide chase. Neurons as in **a** were subjected to a cycloheximide (CHX) chase experiment for the indicated time to stop protein translation. Remaining protein levels were quantified by immunoblotting, normalized to α-tubulin levels. Data are means ± SEM (*$p < 0.05$, **$p < 0.01$, ***$p < 0.001$ by two-way ANOVA; $n = 3$ independent experiments). **c**–**f** Turnover of Munc18-1 by Dendra2 photoconversion. HEK293T cells were transfected with WT or mutant Munc18-1b:Dendra2 fusion constructs. Two days after transfection, expressed Dendra2 was photoconverted. The green signal was quantified before and after photoconversion (**c**, **d**), and the red signal was quantified at 0, 3, and 24 h after photoconversion (**e**, **f**). Data are means ± SEM (*$p < 0.05$, **$p < 0.01$, ***$p < 0.001$ by Student's $t$ test in **d** and **f**, and by two-way ANOVA in **e**; $n = 4$–6 independent experiments). Scale bar in **c** = 50 μm

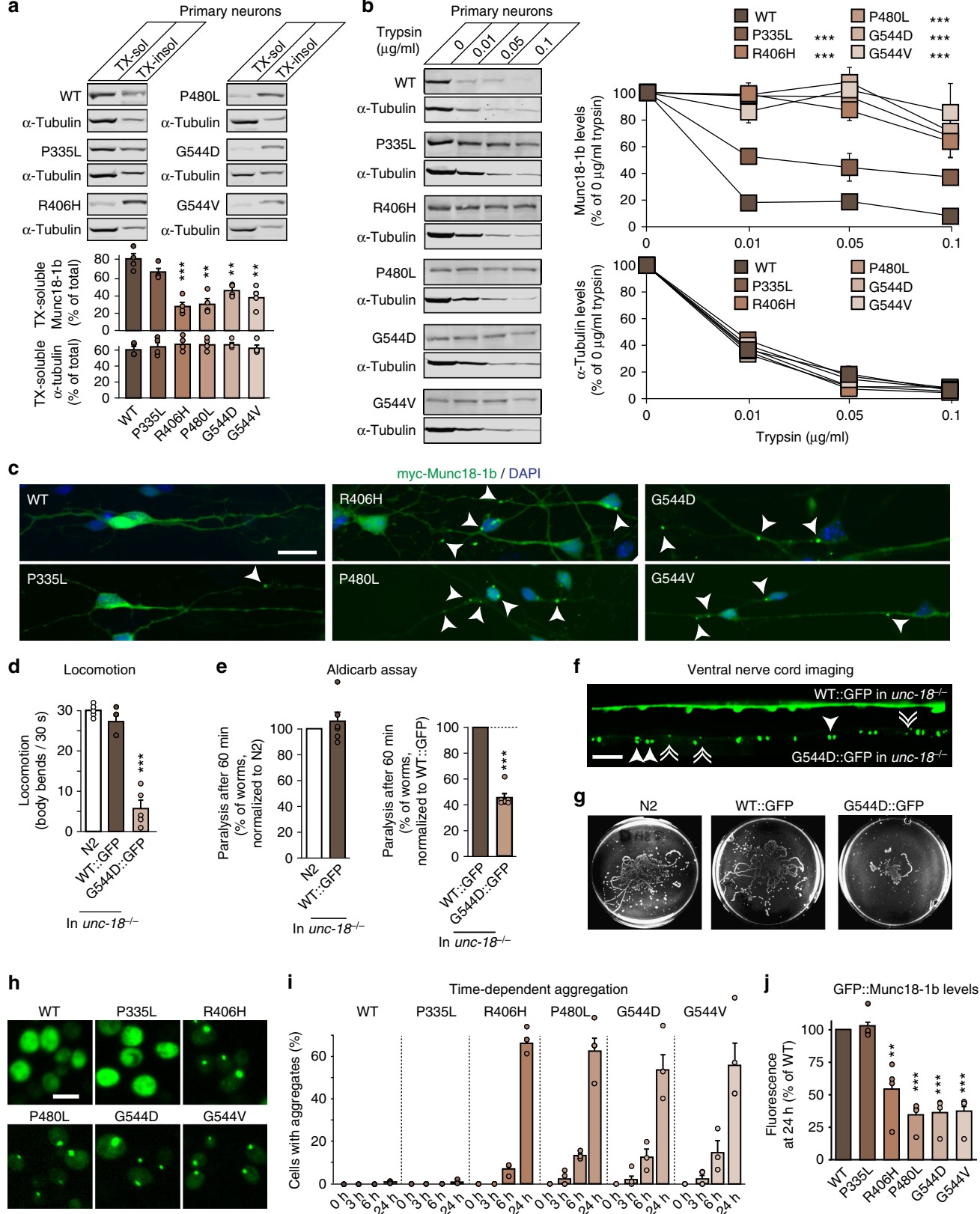

infected primary heterozygous Munc18-1 neurons with lentivirus-expressing GFP-tagged Munc18-1 variants and assessed the levels of the GFP-tagged protein as well as the levels of endogenous WT Munc18-1. We confirmed that the GFP-tagged Munc18-1 mutants are expressed at lower levels than GFP-tagged

WT Munc18-1 (Fig. 5a), likely due to rapid degradation as shown above. Strikingly, this destabilization was accompanied by a reduction in endogenous WT Munc18-1 levels (Fig. 5a). To exclude a potential effect of the GFP-tag on Munc18-1 stability, we co-infected primary Munc18-1 knockout neurons with

**Fig. 4** Aggregation of Munc18-1 mutants. **a** Solubility of Munc18-1. Munc18-1 knockout neurons expressing WT or mutant Munc18-1b were solubilized in 0.1% Triton X-100 (TX). Equal volumes of soluble and insoluble fractions were analyzed by quantitative immunoblotting (α-tubulin = control). Data are means ± SEM (**$p < 0.01$, ***$p < 0.001$ by Student's $t$ test; $n = 4$ independent experiments). **b** Limited proteolysis. Neurons as in **a** were incubated with increasing concentrations of trypsin. Remaining protein levels were analyzed by quantitative immunoblotting (α-tubulin = control). Data are means ± SEM (***$p < 0.001$ by two-way ANOVA; $n = 3$ independent experiments). **c** Aggregation of mutant Munc18-1. Neurons as in **a** were analyzed for the subcellular localization of Munc18-1b by immunocytochemistry. Arrows depict aggregates. Scale bar = 20 μm. **d** Locomotion of *C. elegans* expressing GFP-tagged WT or G544D *unc-18*. Body bends per 30 s were counted. Data are means ± SEM (***$p < 0.001$ by Student's $t$ test; $n = 5$ independent experiments on ten worms per experiment). **e** Paralysis of WT worms (N2) or worms expressing GFP-tagged WT or G544D mutant *unc-18* after 60 min exposure to aldicarb. Data are means ± SEM (***$p < 0.001$ by Student's $t$ test; $n = 6$ independent experiments on ten worms per experiment). **f** Lack of axonal localization of mutant UNC-18. *C. elegans* expressing WT::GFP or G544D::GFP were immobilized, and the ventral nerve cord was imaged (solid arrowheads = pairs of bigger puncta, broken arrowheads = single, smaller puncta. Scale bar = 10 μm). **g** Worm traces after the heat shock assay (Supplementary Fig. 8e). **h, i** Aggregation of mutant Munc18-1 in yeast. *S. cerevisiae* expressing GFP-tagged Munc18-1 variants were imaged 24 h after induction of protein expression (**h**) to quantify aggregation (**i**). Data are means ± SEM ($n = 3$ independent experiments). Scale bar = 5 μm. **j** Expression of mutant Munc18-1 in yeast. Munc18-1 levels were analyzed by measuring GFP fluorescence in a plate reader 24 h post induction. Data are means ± SEM (**$p < 0.01$, ***$p < 0.001$ by Student's $t$ test; $n = 4$ independent experiments)

lentiviral vectors expressing myc-tagged mutant Munc18-1 in addition to HA-tagged WT Munc18-1. Again, mutant Munc18-1 levels were significantly reduced and resulted in a reduction in WT protein levels (Fig. 5b). These data suggest that neurons expressing Munc18-1 mutants contain significantly lower levels of WT Munc18-1 than 50%, leading to functional impairments in Munc18-1 exceeding those of the suggested haploinsufficiency.

What could explain the loss of functional WT Munc18-1 levels? Aggregation of mutant Munc18-1 may induce decreased solubility and aggregation of WT Munc18-1. To test this, we first assessed if mutant Munc18-1 is able to interact with WT Munc18-1, using co-immunoprecipitation experiments. We found that WT Munc18-1 forms at least dimers, and that all mutant variants retained the ability to bind to WT Munc18-1 (Fig. 5c), supporting the possibility of co-aggregation of WT and mutant Munc18-1. To test this directly, we repeated the Triton X-100 solubility assay, but co-expressed WT and mutant Munc18-1 in primary Munc18-1 knockout neurons. As before (Fig. 4), mutant Munc18-1 proteins were significantly less soluble compared to WT Munc18-1 (Fig. 5d). Strikingly, presence of mutant Munc18-1 reduced the solubility of co-expressed WT Munc18-1 by ~25%, in both primary neurons and in Neuro2a cells (Fig. 5d and Supplementary Fig. 10a), suggesting co-aggregation of mutant and WT Munc18-1.

To test this effect of mutant Munc18-1 on aggregation of WT Munc18-1 in an independent assay, we performed limited proteolysis with co-expression of WT and mutant Munc18-1, and found that WT Munc18-1 was significantly less prone to tryptic digestion when co-expressed with mutant Munc18-1 variants (Supplementary Figs. 10b–e). We again did not find a toxic effect of Munc18-1 aggregates under these conditions, as assessed by an MTT assay and nuclear staining of primary neurons (Supplementary Figs. 10f and 10g).

To test this gain-of-toxic function of mutant Munc18-1 on synapse function in vivo, we generated transgenic *C. elegans* expressing WT or mutant *unc-18* on an N2 WT background. At similar expression levels (Supplementary Fig. 11a), worms expressing mutant *unc-18* but not those expressing WT *unc-18* exhibited significant impairments in locomotion, heat shock paralysis and neurotransmitter release (Fig. 5e–g and Supplementary Figs. 11b and 11c).

In summary, our data support a disease model where Munc18-1-linked encephalopathies are not caused by haploinsufficiency, but by a dominant-negative disease mechanism, where instability and aggregation of mutant Munc18-1 triggers similar impairments in WT Munc18-1. Thus, rescue strategies aimed at restoring either mutant and/or WT Munc18-1 levels could be expected to ameliorate these impairments.

**Chemical chaperones augment Munc18-1 levels and solubility.** Multiple studies have demonstrated that chemical chaperones can reverse the mislocalization of proteins associated with various conformational diseases, and slow, arrest, or reverse pathology in in vitro and in vivo models of cystic fibrosis[43], sickle cell disease[44], and neurodegenerative diseases[45–48], among others. Their mode of action is diverse and includes stabilization of improperly folded proteins, reduction of aggregation, prevention of non-productive interactions with other resident proteins, and alteration of the activity of endogenous chaperones in such a way that the affected proteins are more efficiently folded and transported to the appropriate intracellular or extracellular destination. Yet, to our knowledge, these have never been tested or even considered to treat Munc18-1-linked encephalopathies.

We focused on screening chemical chaperones at concentrations that have been successfully used in vitro and in vivo to stabilize protein structures and/or prevent protein aggregation in other diseases, including trimethylamine N-oxide (TMAO), sorbitol, betaine, trehalose, glycerol, 4-phenylbutyrate, and sodium butyrate (Supplementary Fig. 12a). Three chemical chaperones stabilized the levels of mutant and wild-type Munc18-1 in transfected HEK293T cells: 4-phenylbutyrate, sorbitol, and trehalose (Supplementary Figs. 12a and 12b). We independently confirmed the rescue effects of sorbitol and trehalose in our yeast model system (Supplementary Fig. 12c). 4-Phenylbutyrate is absent from these experiments since it was toxic to the yeast cells, dramatically impairing their growth. We then tested the effect of these chemical chaperones on total Munc18-1 levels in primary neurons, and found a significant increase in WT as well as mutant Munc18-1 levels with all three chaperones (Fig. 6a).

Next, we asked whether the three chemical chaperones only augment total Munc18-1 protein levels, or if they are also able to affect the insolubility of mutant Munc18-1. We detected changes in the number of aggregate puncta by immunocytochemistry in primary neurons (Supplementary Fig. 13a). However, imaging aggregates are an unreliable means of measuring protein insolubility. In addition, smaller aggregates may not be visible at all. In fact, solubility of mutant Munc18-1 was severely affected when assessed biochemically, while immunostaining revealed only a small fraction of aggregated Munc18-1 (compare Fig. 4a, b with Fig. 4c). We therefore assessed protein solubility biochemically, using a Triton X-100 solubility assay, which enables detection of aggregates irrespective of their size. We found a significant improvement of mutant Munc18-1 solubility in primary neurons with all three chaperones (Fig. 6b), but not of PSD-93 (Supplementary Fig. 13b), suggesting that 4-phenylbutyrate, sorbitol, and trehalose not only increase total Munc18-1

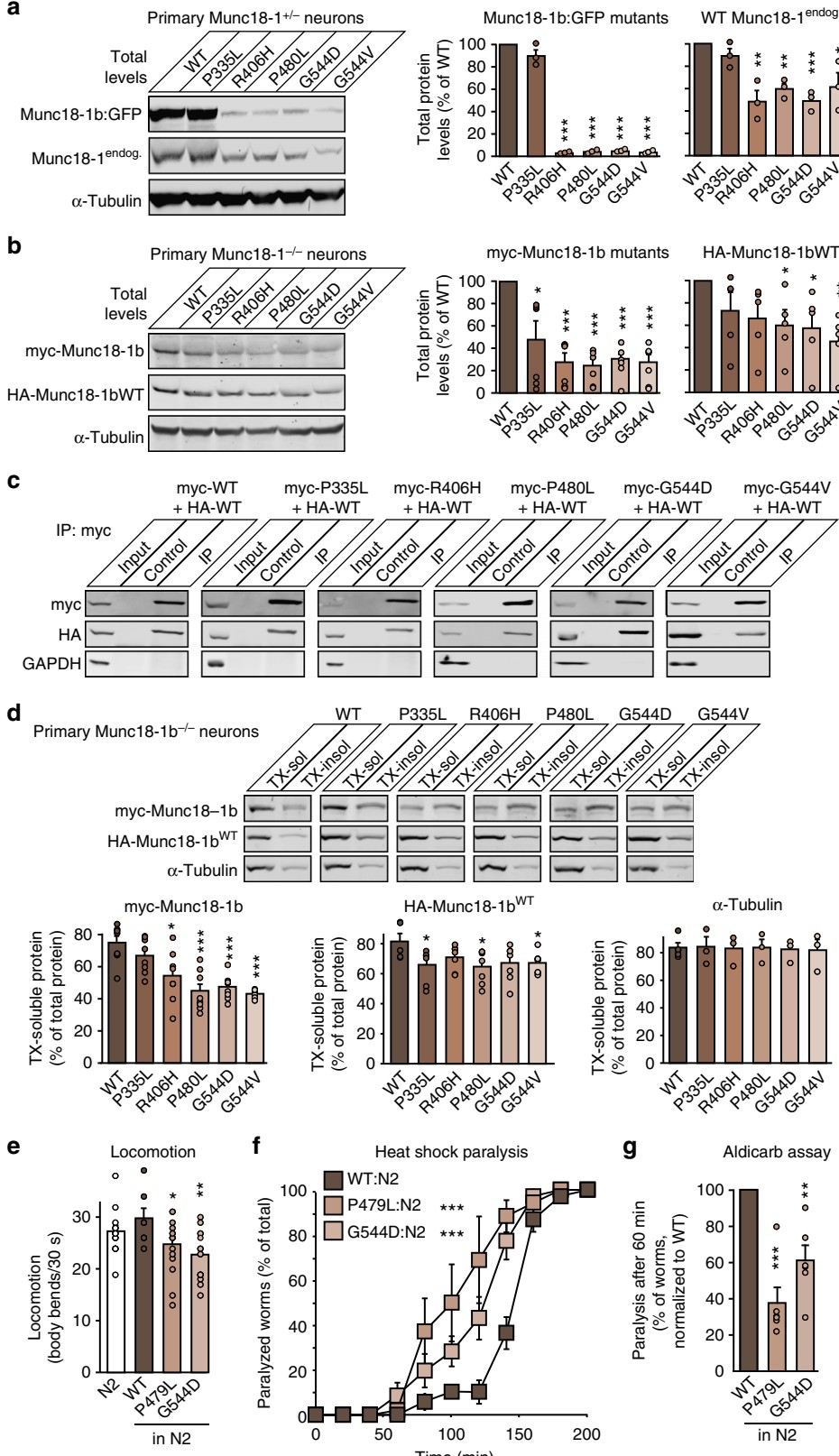

levels, but also rescue the insolubility of mutant Munc18-1, thereby further boosting the soluble functional pool of Munc18-1. Importantly, even massive overexpression of Munc18-1 does not have a deleterious effect on exocytosis[49–53], enabling this overall increase of functional Munc18-1 levels as therapeutic strategy.

**Chemical chaperones restore synaptic deficits.** Does rescue of Munc18-1 levels and solubility restore the synaptic deficits we have described above? To test this, we repeated the antibody uptake assay in primary neurons in presence of the chaperones. We found 4-phenylbutyrate, sorbitol, and trehalose to

**Fig. 5** Dominant-negative activity of mutant Munc18-1 on wild-type Munc18-1. **a** Protein levels of Munc18-1 in primary neurons. Levels of GFP-tagged WT or mutant Munc18-1 and endogenous Munc18-1 in heterozygous Munc18-1 neurons were analyzed by quantitative immunoblotting, normalized to α-tubulin levels. Data are means ± SEM (*$p < 0.05$, **$p < 0.01$, ***$p < 0.001$ by Student's $t$ test; $n = 3$ independent experiments). **b** Same as in **a**, except that myc-tagged Munc18-1 variants were co-expressed with HA-tagged WT Munc18-1 in Munc18-1 knockout neurons. Data are means ± SEM (*$p < 0.05$, **$p < 0.01$, ***$p < 0.001$ by Student's $t$ test; $n = 5$–6 independent experiments). **c** Co-immunoprecipitation of mutant with wild-type Munc18-1. Lysates of HEK293T cells that were co-transfected with HA-tagged WT and myc-tagged mutant Munc18-1 were subjected to immunoprecipitation with an anti-c-myc antibody (IP) or no antibody (control). Precipitated myc- and HA-tagged Munc18-1 was analyzed with the respective input by immunoblotting. GAPDH served as control. **d** Solubility of Munc18-1. Primary neurons infected as in **b** were solubilized in 0.1% Triton X-100 (TX). Equal volumes of TX-soluble and -insoluble fractions were analyzed by quantitative immunoblotting. α-Tubulin served as control. Data are means ± SEM (*$p < 0.05$, ***$p < 0.001$ by Student's $t$ test; $n = 5$–7 independent experiments). **e** Locomotion of *C. elegans*. Body bends per 30 s were counted. Data are means ± SEM (*$p < 0.05$, **$p < 0.01$ by Student's $t$ test; $n = 15$ independent experiments on ten worms per experiment). **f** Heat-induced paralysis. Paralysis at 37 °C was scored at indicated time points. Data are means ± SEM (***$p < 0.001$ by two-way ANOVA, compared to N2; $n = 3$ independent experiments on ten worms per experiment). **g** Mutants display reduced acetylcholine release at the worm neuromuscular junction. Paralysis of N2 worms expressing WT or mutant unc-18 was measured after 60 min exposure to aldicarb. Data are means ± SEM (**$p < 0.01$, ***$p < 0.001$ by Student's $t$ test; $n = 6$ independent experiments on 20 worms per experiment)

---

significantly rescue the deficits in neurotransmitter release associated with mutations in Munc18-1, restoring synaptic function back to WT levels (Fig. 6c, d and Supplementary Fig. 14).

We next assessed the effect of chemical chaperones on neuron function in vivo, in both our loss-of-function and gain-of-toxic-function transgenic *C. elegans* strains. 4-Phenylbutyrate, sorbitol, and trehalose rescued the locomotion impairments (Fig. 7a and Supplementary Figs. 15a and 15b), restored the defects in neurotransmitter release (Fig. 7b and Supplementary Figs. 15c, 15d and 17a), and delayed paralysis during heat shock in both worm models (Fig. 7c and Supplementary Fig. 16). Similarly, the three chaperones rescued locomotion, neurotransmitter release, and accelerated heat shock paralysis in our CRISPR-generated P334L and R405H knock-in worms (Fig. 7d, e and Supplementary Figs. S17b and 17c).

In addition, 4-phenylbutyrate, sorbitol, and trehalose strikingly reduced the formation of aggregates in the ventral nerve cord of *C. elegans* expressing GFP-tagged mutant UNC-18, restoring the typical localization within the ventral nerve cord to that of wild-type UNC-18 (Fig. 7f), which was accompanied by ameliorating the defects in neurotransmitter release in these worms (Fig. 7g and Supplementary Figs. 17d and 17e).

## Discussion

The treatment options for Munc18-1-linked encephalopathies can be summarized as frustratingly difficult. Due to a lack of understanding of the underlying disease mechanism, current treatment regimens focus on mitigating disease symptoms and improving day-to-day living, rather than on the specific disease causes. Treatment of Ohtahara syndrome with anti-epileptic drugs is usually ineffective, and the prognosis is very poor, with 50% of infants dying within weeks or months after disease onset[14]. West syndrome is treated commonly with the GABA analog vigabatrin and steroids, and has a poor prognosis, with only 15–30% of cases becoming seizure-free and developing normally or near normally[14]. The response of patients with Dravet syndrome to anti-epileptic drugs is poor, and certain drugs such as the sodium channel blockers carbamazepine, phenytoin, and lamotrigine even exacerbate seizures[14]. Likewise, medical management of Rett syndrome, as well as of ataxia, tremor, and neurodegeneration in patients with Munc18-1 mutations is challenging. Thus, understanding of the disease mechanism is essential to develop rational therapeutic strategies.

Combining molecular, cell biological, and whole animal approaches, we have made six principal observations: (1) At least five missense mutations in Munc18-1 that are linked to varied encephalopathies result in enhanced turnover of the mutant protein (Fig. 3). (2) Remaining mutant protein that cannot be cleared by the cell readily aggregates (Fig. 4). (3) Insolubility of mutant Munc18-1 renders WT Munc18-1 more insoluble and aggregation-prone, reducing functional Munc18-1 levels significantly below 50% (Fig. 5). (4) These impairments result in deficits in synapse function (Figs. 1, 2, 4, and 5). (5) The three chemical chaperones 4-phenylbutyrate, sorbitol, and trehalose stabilize functional Munc18-1 levels and rescue insolubility of WT and mutant Munc18-1 (Fig. 6). (6) Restoring Munc18-1 levels and solubility with chemical chaperones rescues synaptic deficits and reduces aggregation (Figs. 6 and 7). This rescue strategy is not only predicted to work for all identified missense mutations, but also for nonsense and truncations mutants, due to stabilization of the remaining WT Munc18-1 protein in affected patients.

Our study provides for the first time a clear and comprehensive understanding of Munc18-1-linked disease mechanisms where encephalopathies are not caused by haploinsufficiency, but rather are a result of a dominant-negative disease mechanism, impairing neuron function more than haploinsufficiency would entail (Fig. 8). Importantly, our study provides a targeted rescue strategy to overcome the deficits caused by loss of function of WT and mutant Munc18-1, restoring functional Munc18-1 levels with the three chemical chaperones 4-phenylbutyrate, sorbitol, and trehalose (Fig. 8).

Interestingly, the P335L mutation revealed a milder phenotype compared to the other mutations. P335 is located within domain 3a of Munc18-1 (Supplementary Fig. 1b), a region that undergoes conformational changes to mediate binding to syntaxin-1, VAMP2, or the assembled SNARE complex[31–33]. In line with our data showing that P335L enhances neurotransmitter release, P335A results in an extended alpha-helical conformation and accelerates lipid mixing[54]. With this requirement for flexibility surrounding residue P335, mutations in this domain are expected to be less deleterious compared to mutations in other domains of Munc18-1, and this is likely the reason for the intermediate-level deficits that we have observed. In addition, our assays have not demonstrated a consistent dominant-negative mode of action of P335L on WT Munc18-1, classifying this mutant as a loss-of-function mutation in the sense that the ability to undergo conformational changes in domain 3a is lost.

How do 4-phenylbutyrate, sorbitol, and trehalose rescue Munc18-1-linked neuronal dysfunction? Osmolyte chaperones such as sorbitol and trehalose work through altering solvent properties[55], resulting in protein stabilization[56,57], in inhibition of unfolding of native conformations[58], and in reduction of aggregation of proteins that have already misfolded[59]. In addition, osmolyte chaperones have been shown to modulate the function of molecular chaperones[60], thereby improving the efficiency of the protein quality-control system. In contrast, the mechanism of

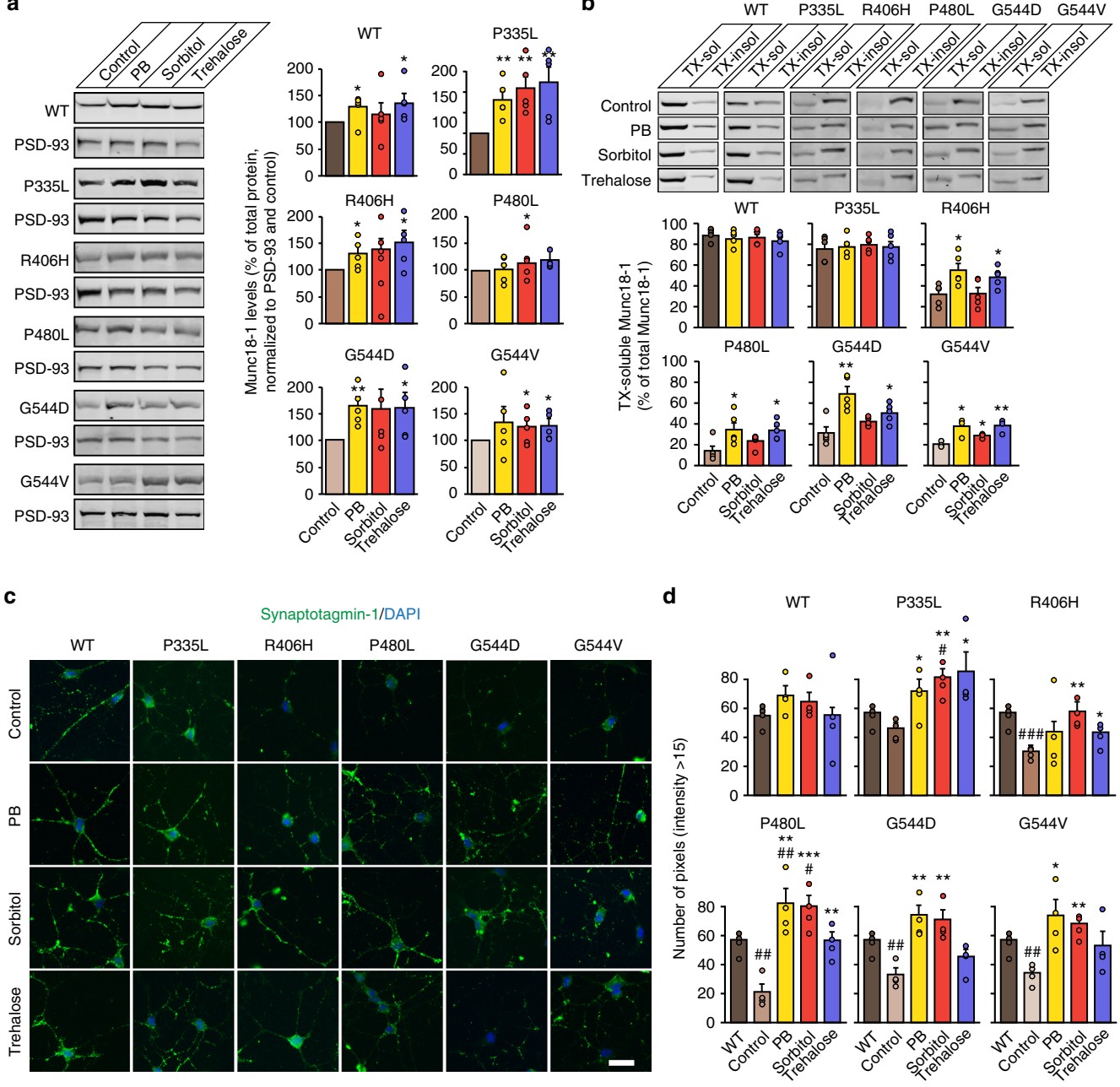

**Fig. 6** Chemical chaperones rescue mutant Munc18-1 deficits in neurons. **a** Total protein levels of Munc18-1. WT and mutant Munc18-1b were expressed in primary cortical neurons infected with lentiviral vectors expressing cre recombinase and Munc18-1b variants in the presence or absence of chemical chaperones. Total protein levels were quantified 7 days after infection by immunoblotting, normalized to the levels of the post-synaptic protein PSD-93. Data are means ± SEM (*$p < 0.05$, **$p < 0.01$ by Student's $t$ test; $n = 5$ independent experiments). **b** Triton X-100 solubility of Munc18-1. WT or mutant Munc18-1b were expressed as above. Seven days after infection, cells were solubilized in 0.1% Triton X-100 (TX). Equal volumes of soluble and insoluble fractions were separated by SDS-PAGE, and TX-soluble Munc18-1 was measured as percent of total Munc18-1 by quantitative immunoblotting. Solubility of PSD-93 served as control (Supplementary Fig. 13b). Data are means ± SEM (*$p < 0.05$, **$p < 0.01$ by Student's $t$ test; $n = 5$ independent experiments). **c**, **d** Uptake of synaptotagmin-1 antibody during high K$^+$ stimulation. Primary cortical neurons infected at 6 DIV with lentivirus expressing cre recombinase and WT or mutant Munc18-1b were subjected to an antibody uptake assay at 13 DIV in the absence or presence of chemical chaperones. Endocytosed synaptotagmin-1 antibody was quantified by immunostaining (**c**), via counting the number of pixels > intensity of 15 (**d**). Data are means ± SEM (*,#$p < 0.05$, **,##$p < 0.01$, ***,###$p < 0.001$ by Student's $t$ test; $n = 4$ independent experiments). Scale bar in **c** = 20 µm

action for hydrophobic chaperones, such as 4-phenylbutyrate, involves the interaction of hydrophobic regions of the chaperone with the exposed hydrophobic segments of the unfolded protein, which protects the protein from aggregation (reviewed in ref. [61]).

Importantly, 4-phenylbutyrate and trehalose cross the blood–brain barrier effectively, and previous studies have demonstrated beneficial effects of these chaperones at the concentrations used in this study on survival in mouse models of

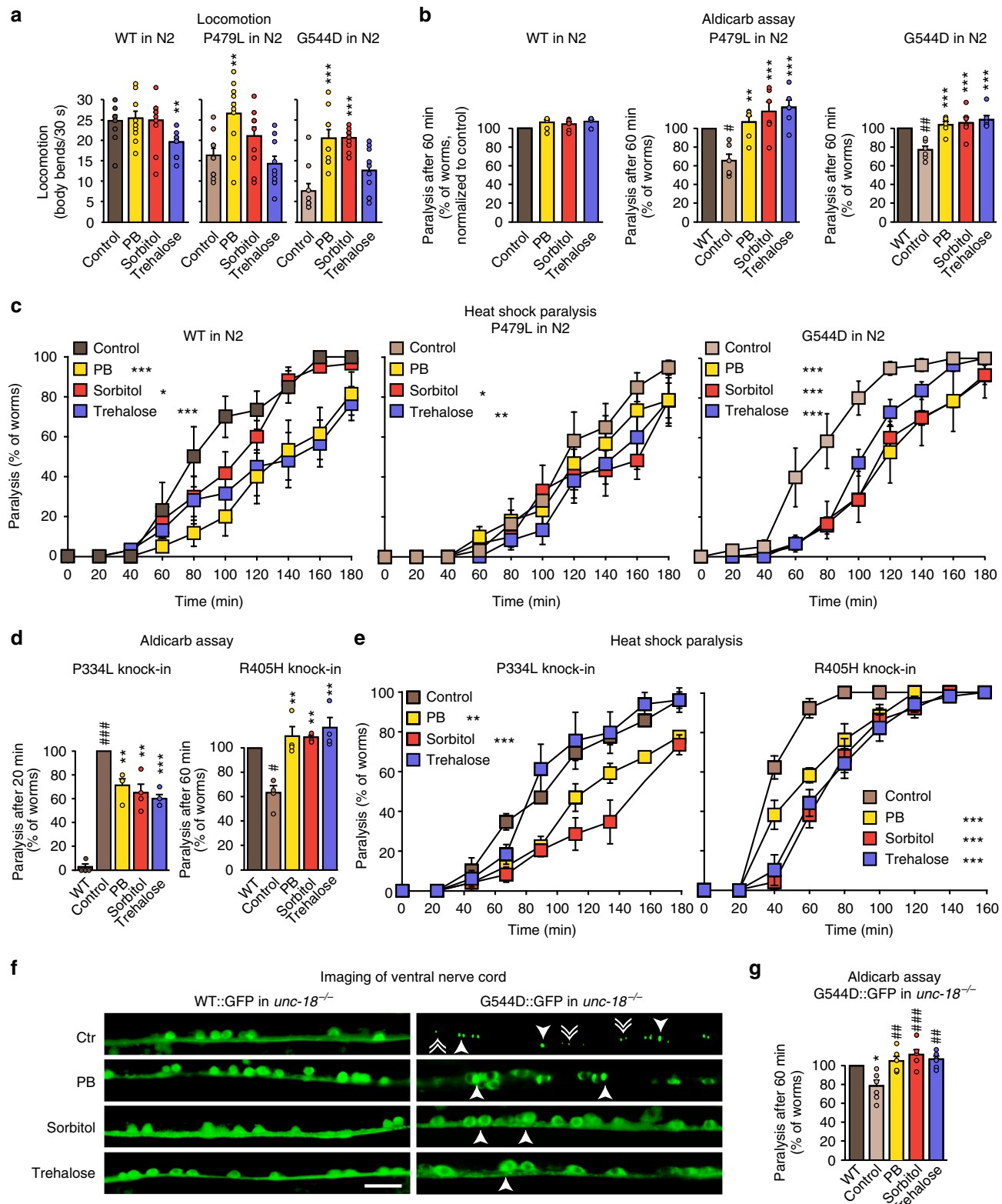

Parkinson's, Alzheimer's, and Huntington's disease[47,62–65]. In addition, 4-phenylbutyrate is orally bioavailable, FDA-approved for treatment of urea cycle disorders[66], and is in clinical trial for the treatment of cystic fibrosis[67].

A common criticism for the usage of chemical chaperones is the need for a high concentration and their lack of specificity.

Thus, identifying pharmacological chaperones, which are more specific at much lower concentration and are thus expected to have less impact on overall cellular processes, may be a logical next step. Yet, it is apparent from the data presented here that Munc18-1-linked encephalopathies are prime candidates for testing chemical chaperone therapies.

**Fig. 7** Chemical chaperones rescue deficits of mutant UNC-18 in *C. elegans*. **a** Locomotion of *C. elegans*. Body bends per 30 s were counted. Data are means ± SEM (**$p < 0.01$, ***$p < 0.001$ by Student's *t* test; $n = 10$ independent experiments on ten worms per experiment). **b** Rescue of reduced acetylcholine release in worms expressing mutant UNC-18 variants. Young adult worms expressing WT or mutant *unc-18* were exposed to aldicarb, and paralysis at 60 min was measured. Data are means ± SEM (#$p < 0.05$, **,#$p < 0.01$, ***$p < 0.001$, by Student's *t* test; $n = 6$ independent experiments on 20 worms per experiment). **c** Heat-induced paralysis. Indicated worm strains were exposed to 37 °C over a period of 180 min, and paralysis was scored at indicated time points. Data are means ± SEM (*$p < 0.05$, **$p < 0.01$, ***$p < 0.001$ by two-way ANOVA, compared to control, $n = 5$–6 independent experiments on ten worms per experiment). **d**, **e** Rescue of reduced acetylcholine release and heat-induced paralysis in CRISPR/Cas9-generated P334L and R405H knock-in worms. Worms were analyzed as described in **a**–**c**. Data are means ± SEM (*$p < 0.05$, **$p < 0.01$, ***$p < 0.001$ by two-way ANOVA, compared to control, $n = 4$ independent experiments on ten worms per experiment for **d**, and $n = 4$ independent experiments on 20 worms per experiments for **e**). **f** Rescue of subcellular localization of UNC-18 in worms expressing mutant UNC-18. *C. elegans* expressing WT::GFP or G544D::GFP were immobilized, and the ventral nerve cord was imaged. Solid arrowheads point to pairs of bigger puncta, broken arrowheads to single, smaller puncta. Scale bar = 10 μm. **g** Rescue of reduced acetylcholine release in worms expressing GFP-tagged mutant UNC-18 variants. Experiments were performed as described under (**b**). Data are means ± SEM (*$p < 0.05$, ##$p < 0.01$, ###$p < 0.001$, by Student's *t* test; $n = 6$ independent experiments on 20 worms per experiment)

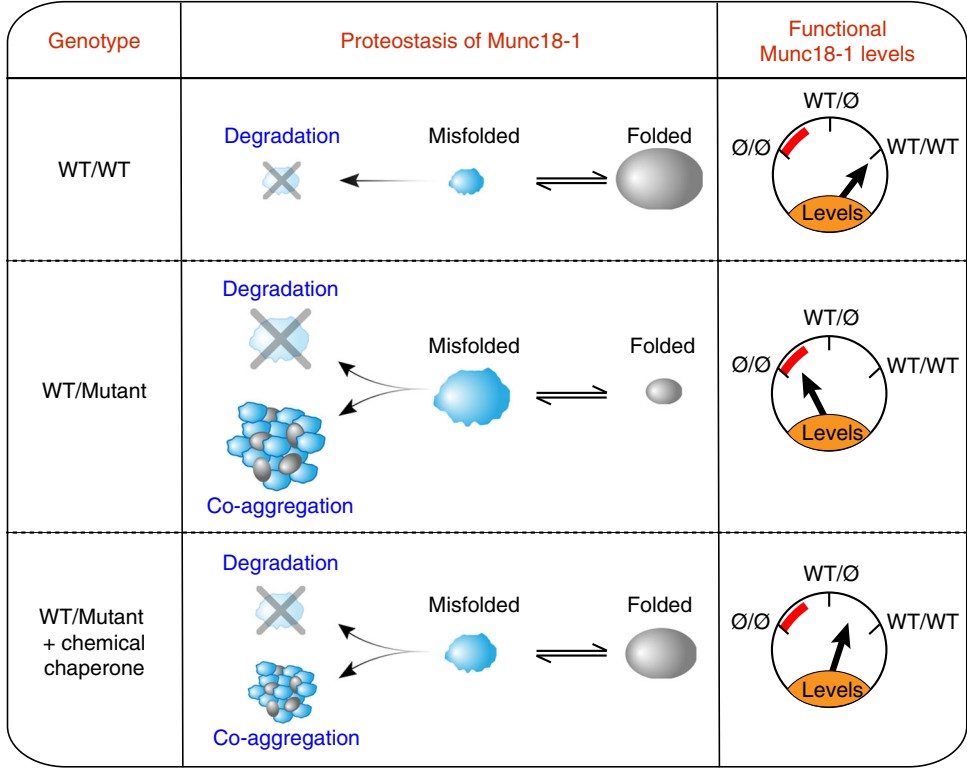

**Fig. 8** Model of Munc18-1 dysfunction in encephalopathies and rescue of deficits with chemical chaperones. Munc18-1-linked encephalopathies are caused by a dominant-negative disease mechanism. Heterozygous missense mutations in Munc18-1 cause a reduction in functional Munc18-1 levels significantly below 50%, due to accelerated degradation of misfolded mutant Munc18-1, aggregation of misfolded mutant Munc18-1 that is resistant to cellular clearance, and due to co-aggregation of WT Munc18-1. Chemical chaperones not only shift the unfolded–folded protein equilibrium significantly toward a folded state, but also result in an increase in total Munc18-1 levels. This overall increase in Munc18-1 levels and solubility is sufficient to rescue the Munc18-1-linked neuronal deficits in vitro and in vivo

## Methods

**Mouse strains**. The conditional Munc18-1 knockout mouse was kindly donated by Dr. Matthijs Verhage at CNCR (the Netherlands). Mice were bred either as homozygous or heterozygous conditional Munc18-1 knockout mice. No inclusion criteria for neuronal cultures were used. Mice were housed with a 12-h light/dark cycle in a temperature-controlled room with free access to water and food. All animal procedures were performed according to NIH guidelines and approved by the Committee on Animal Care at Weill Cornell Medicine.

***C. elegans* strains and maintenance**. Strains were generated by standard methods. For generation of transgenic worms, plasmids encoding the UNC-18 variants under a neuronal promoter (*Psnb-1*) were injected into either N2 or *unc-18*(e81) along with a fluorescent marker to identify transgenic progeny. At least two stable lines were identified for each injection. For generation of CRISPR/Cas9-edited worms, crRNAs targeting the *unc-18* gene were selected using the www.crispr.mit. edu design tool (5'-UUCAACUCUUUCUUGUGCUG-3' for P334L, and 5'-UGUACAACAGAAUCAAUCUG-3' for R405H). As a positive control for Cas9 activity, crRNA targeting the *dpy-10* gene for the *cn64* allele was used[68]. Annealed gRNAs were generated by mixing crRNAs with tracrRNA at a 2:1 ratio according to the manufacturer's protocol (Synthego). ssODN repair templates contained the desired mutation, silent mutations to the PAM site, and the sequence homologous to the gRNA to prevent re-editing, a restriction enzyme recognition site to facilitate screening for positive integration events, and 45 base-long flanking homology arms (5'-GACTCCAAATCGATCAAGGATCTATCGATGCTCATCAAAAGAATGCT GCAGCATAAAAAGGAATTAAACAAGTTTAGCACTCACATCAGTCTTGC TGAGGAATGCATGAAA-3' for P334L; 5'-GATGGTGCCACTTTTGATTGA CCCAGCCGTGCGGGTGTGAAGACCGTCTTCACTTGATTCTGTTGTACATT CTTTCCAAGAATGGAATTACTGATGAA-3' for R405H). N2 animals were injected with a mix of 5 μM *unc-18* gRNA, 5 μM *dpy-10* gRNA, 2 μM *unc-18*

ssODN, 2 μM *dpy-10* ssODN, and 5 μM Cas9 protein in 300 mM KCl and 20 mM HEPES pH 7.4. 100–200 F1 dpy animals were singled for each injection. For 100–200 F2 offspring plates, 50 animals were lysed for 60 min at 60 °C (lysis buffer: 50 mM KCl, 2.5 mM MgCl$_2$, 0.45% Tween-20, 0.45% IGEPAL, 1% proteinase K in 10 mM Tris, pH 8.0). Proteinase K was subsequently deactivated by a 15-min incubation at 95 °C. Worm lysates were subjected to PCR (5′-CACTCCATACG AAAGCCAGGTGAG-3′ and 5′-CCCAAGTAAGCAGCGTTGGTGA-3′ for P334L; 5′-CTCGACGCTTACAAGGCTGACG-3′ and 5′-CTGGAACCCAGC GGGAAGAT-3′ for R405H), and the PCR products were digested with the respective restriction enzyme introduced with the repair template, and analyzed on an agarose gel. The presence of the desired mutations was further confirmed by sequencing. Each worm strain was subsequently outcrossed to N2 worms four times. All animals were maintained on agar nematode growth media at 20 °C and seeded with OP50 bacteria.

**C. elegans locomotion assay**. Ten to twenty young adult worms were placed on agar plates and gently moved to the microscope to not mechanically stimulate the worms. Body bends were counted for 30 s upon stimulating the end of each worm with a platinum/iridium wire. Each genotype was coded and tested blindly.

**Aldicarb assay**. To measure aldicarb sensitivity, 20–25 young adult worms were placed on agar plates containing 1 mM aldicarb. Worms were scored for paralysis every 10 min by recording body movement in response to prodding the tail with a platinum/iridium wire. Each genotype was coded, tested blindly, and paralysis curves were generated by averaging individual curves.

**C. elegans imaging**. Animals were immobilized using a 30 mg/ml solution of 2,3-butanedione monoxime in M9 buffer (22 mM KH$_2$PO$_4$, 42.3 mM Na$_2$HPO$_4$, 85.6 mM NaCl, and 1 mM MgSO$_4$), were then mounted on 2% agarose pads, and the ventral nerve cord was imaged on an Eclipse 80i upright fluorescence microscope (Nikon).

**C. elegans heat shock paralysis**. Ten young adult animals were placed on agar plates and subjected to 37 °C in an incubator. All animals were confirmed to be capable of locomotion prior to the assay by observing their movement in response to a head-poke stimulus. At each time point, the percentage of paralyzed animals was assessed by their ability to move in response to a head-poke stimulus. Each genotype was coded, tested ~10 times blindly, and the paralysis curves were generated by averaging paralysis time courses for each plate.

**Cell culture and maintenance**. HEK293T and Neuro2a cells (both from ATCC) were maintained in DMEM with 1% penicillin/streptomycin and 10% bovine serum. Cells were solely used as protein expression systems or to produce lentivirus, and have not been authenticated or tested for mycoplasma contamination. Mouse cortical neurons were cultured from newborn mice of either sex. Cortices were dissected in ice-cold HBSS, dissociated and triturated with a siliconized pipette, and plated onto 6 mm poly L-lysine-coated coverslips (for immunofluorescence) or on 24-well plastic dishes. Plating media (MEM supplemented with 5 g/l glucose, 0.2 g/l NaHCO$_3$, 0.1 g/l transferrin, 0.25 g/l insulin, 0.3 g/l L-glutamine, and 10% fetal bovine serum) was replaced with growth media (MEM containing 5 g/l glucose, 0.2 g/l NaHCO$_3$, 0.1 g/l transferrin, 0.3 g/l L-glutamine, 5% fetal bovine serum, 2% B-27 supplement, and 2 μM cytosine arabinoside) 2 days after plating. At 6 days in vitro (DIV), neurons were transduced with recombinant lentiviruses expressing GFP-tagged, myc-tagged, or HA-tagged Munc18 variants, cre recombinase, and/or Δcre (as control). Neurons were harvested or used for experiments as indicated at 13 DIV.

**Neuronal activity**. Primary neurons were plated in 48-well CytoView MEA plates (Axion BioSystems) at a density of 160,000 cells per 10 μl containing 20 μg/ml laminin and transduced as above. Spontaneous activity was recorded at 13 DIV for 10 min using the Maestro Pro MEA System (Axion BioSystems). Extracellular voltage was recorded at each electrode with a sampling rate of 12.5 kHz. Spikes were identified from the raw signals using a detection threshold set independently for each electrode of ±6× the standard deviation of the noise (AxIS Navigator software, Axion Biosystems). Electrodes with at least five spikes/min were considered active electrodes. One well with no active electrodes was excluded from analysis. The Neural Metric Tool (Axion BioSystems) was used for burst and synchrony analysis. Bursts were identified when there were a minimum of five spikes on a single electrode with maximum inter-spike interval of 0.1 s. Network bursts were identified using an adaptive algorithm[69], requiring a minimum of 40 spikes across at least 25% of the electrodes in a well.

**Expression vectors**. Full-length human Munc18-1b cDNA was inserted into pCMV5 or lentiviral vector FUW, containing an N-terminal HA- or myc-tag and a two amino acid linker, resulting in the following N-terminal sequence (EQKLI-SEEDL-GG for myc; YPYDVPDYA-GG for HA). For generation of GFP-tagged Munc18-1b, Emerald was fused C-terminally to Munc18-1b with a 12 amino acid linker (AAAGGSGGSGGS). The same strategy was applied for generation of

Dendra2-tagged Munc18-1b. For generation of transgenic *C. elegans* strains, untagged or C-terminally GFP-tagged *unc-18a* with an AAAGGSGGSGGS linker was inserted into an expression vector carrying a Pnsb-1 promoter, to drive pan-neuronal expression. For generation of transgenic *S. cerevisiae* strains, untagged or GFP-tagged Munc18-1b with an AAAGGSGGSGGS linker was inserted into pYES2 NT/A (Thermo Fisher Scientific). Mutant Munc18-1b or *unc-18a* constructs were generated by site-specific mutagenesis, according to the protocol of the manufacturer (Stratagene).

**Transfection of HEK293T and Neuro2a neuroblastoma cells**. Cells were transfected with cDNA using calcium phosphate produced in house: 1 h prior to transfection, 25 μM chloroquine in fresh media was added. DNA was incubated for 1 min at room temperature in 100 mM CaCl$_2$ and 1× HBS (25 mM HEPES pH 7.05, 140 mM NaCl, and 0.75 mM Na$_2$HPO$_4$) and the transfection mix was then slowly added to the cells. Medium was replaced with fresh medium after 6 h. Cells were harvested or used for experiments as indicated 2 days after transfection. For lentivirus production, HEK293T cells were transfected with equimolar amounts of lentiviral vector FUW containing myc-tagged, HA-tagged, or GFP-tagged Munc18-1, pMD2-G-VSVg, pMDLg/pRRE, and pRSV-Rev. Medium containing the viral particles were collected 48 h later and centrifuged for 10 min at 2000 rpm to remove cellular debris. Viral particles were subsequently concentrated tenfold by centrifugation.

**RNA isolation and qPCR**. RNA was isolated from primary neurons expressing Munc18-1 variants using RNeasy Mini kit according to the manufacturer's protocol (Qiagen). Fifty nanograms of RNA were transcribed into cDNA with integrated removal of genomic DNA contamination using oligo(dT) and random primers (QuantiTect Reverse Transcription kit, Qiagen). qPCR was performed on a CFX96 Real-Time PCR Detection System (BioRad) using the QuantiTect SYBR Green PCR kit (Qiagen) and primers for myc-tagged Munc18-1 (5′-AGAAGCTGATCA GCGAGGAGGAC-3′ and 5′-CTCATGCTTAACTGGTCCACCACCA-3′, resulting in a 145 bp product) and GAPDH as reference (5′-ATGTGTCCGTCGTGG ATCTGACG-3′ and 5′-AAGTCGCAGGAGACAACCTGGTC-3′, resulting in a 142 bp product). Specificity of the PCR reaction was confirmed by melting point analysis and agarose gel electrophoresis. PCR efficiency of both primer pairs was determined by qPCR of tenfold dilutions of cDNA samples and linear regression of Ct values in the range of Ct = 10–20. Upon confirmation of comparable primer efficiency, quantification of mRNA levels was performed using the ΔΔCt method.

**Total protein levels**. Forty-eight hours after transfection of HEK293T or Neuro2a neuroblastoma cells or 7 days after transduction of primary neurons, cells were washed twice in PBS containing 1 mM MgCl$_2$ and solubilized in 2× Laemmli sample buffer containing 77 mg/ml DTT. Samples were then sonicated and boiled for 10 min at 100 °C before separation by SDS-PAGE. For measuring total protein levels in *C. elegans*, worms were removed with 1 ml M9 buffer from a starved 6 cm dish, pelleted by centrifugation, and immediately flash frozen in liquid nitrogen. Worms were thawed in the presence of 2× Laemmli sample buffer containing 77 mg/ml DTT, immediately sonicated and boiled for 10 min at 100 °C.

**Triton X-100 solubilization assay**. Cells were washed twice with PBS containing 1 mM MgCl$_2$ and removed from the dish using PBS. Cells were pelleted by centrifugation (5 min at 500$_{av}$) and were solubilized in 0.1% Triton X-100 in PBS supplemented with protease inhibitors for 1 h at 4 °C under constant agitation. Insoluble material was pelleted by centrifugation (10 min at 13,000$_{av}$ and 4 °C), the Triton X-100 soluble supernatant was transferred to a fresh tube, and the pellet was adjusted to the same volume with PBS. Both fractions were supplemented with 5× Laemmli sample buffer containing 77 mg/ml DTT and sonicated before separating equal volumes by SDS-PAGE.

**Limited proteolysis**. Cells were washed twice in PBS containing 1 mM MgCl$_2$, lysed by freezing for at least 60 min at −80 °C, and incubated in 0.01, 0.05, 0.10, 0.50, or 1.0 μg/μl trypsin on ice for 5 min. Tryptic digestion was immediately stopped by addition of 5× Laemmli sample buffer containing 77 mg/ml DTT and boiling for 10 min at 100 °C.

**Cycloheximide chase experiments**. Cells were incubated in 100 μg/ml cycloheximide for 0, 3, 8, or 24 h at 37 °C and 5% CO$_2$. Cells were then washed twice in PBS containing 1 mM MgCl$_2$, and were resuspended in 2× Laemmli sample buffer containing 77 mg/ml DTT.

**Dendra2 photoconversion experiments**. Munc18-1b:Dendra2 fusion proteins expressed in HEK293T cells or primary neurons were photoconverted on an EVOS FL inverted fluorescence microscope (Life Technologies) from a green to a red fluorescent form using a 1 min exposure to UV light. Degradation of Munc18-1b variants in HEK293T cells was quantified by time-lapse imaging of the activated signal at the single-cell level. Degradation in primary neurons was analyzed by

imaging of the activated signal in neuronal processes of fixed neurons on an Eclipse 80i upright fluorescence microscope (Nikon).

**Immunocytochemistry**. Cells were washed twice with phosphate-buffered saline (PBS) containing 1 mM $MgCl_2$ and were fixed with 4% paraformaldehyde in PBS for 20 min at room temperature. Cells were washed twice with PBS and permeabilized with 0.1% Triton X-100 in PBS for 5 min at RT. After washing twice with PBS, cells were blocked for 20 min with 5% bovine serum albumin (BSA) in PBS. Primary antibody was added in 1% BSA in PBS over night at 4 °C. The next day, cells were washed twice in PBS, blocked for 20 min in 5% BSA in PBS and incubated with secondary antibody and DAPI in 1% BSA in PBS for 1 h at RT in the dark. Cells were washed twice with PBS and were mounted using Fluoromount-G. Cells were imaged on an Eclipse 80i upright fluorescence microscope (Nikon).

**Metabolic activity**. Metabolic activity of primary neurons and Neuro2a cells was assessed using an MTT assay. Forty-eight hours after transfection of Neuro2a or 7 days after infection of primary neurons, medium was changed to fresh medium containing 0.5 mg/ml MTT and cells were incubated for 1 h at 37 °C. Medium was then removed and 200 μl of isopropanol/1 M HCl (ratio 24:1) was added to the cells. After 2 min incubation, the absorbance at 560 nm was measured in a plate reader (Synergy H1 Hybrid Reader, BioTek), subtracted by the absorbance at 620 nm (reference wavelength).

**Antibody uptake assay**. Cells were equilibrated for 10 min at room temperature in Krebs–Ringer solution (128 mM NaCl, 25 mM HEPES, 4.8 mM KCl, 1.3 mM $CaCl_2$, 1.2 mM $MgSO_4$, 1.2 mM $KH_2/K_2HPO_4$ [pH 7.4], 5.6% glucose, pH 7.4). Medium was then replaced with Krebs–Ringer solution containing 55 mM KCl (and a corresponding reduction in NaCl) or with regular Krebs–Ringer solution (low $K^+$ control) for 20 min at room temperature, containing 1:50 dilution of lumenal synaptotagmin-1 antibody. Cells were washed three times for 1 min with Krebs–Ringer solution, and were then fixed with 4% PFA in PBS, and incubated with secondary antibody and DAPI as described above. Neurons were imaged on an Eclipse 80i upright fluorescence microscope (Nikon) at same fluorescence intensity settings. For each image, distribution of the intensity of 300 pixels of dendrites and axons was analyzed, subtracted by average pixel intensity of the background, using ImageJ (NIH). Data were grouped into bins of five and number of pixels in each pixel intensity group was plotted.

**Quantitative immunoblotting**. Protein samples were separated by SDS-PAGE and transferred onto nitrocellulose membranes. Blots were blocked in Tris-buffered saline (TBS) containing 0.1% Tween-20 (TBS-T) containing 5% fat-free milk for 30 min at room temperature. The blocked membrane was incubated overnight in PBS containing 1% BSA and 0.2% $NaN_3$ and the primary antibody. The blots were then washed twice in TBS-T containing 5% fat-free milk, then incubated for 1 h in the same buffer containing secondary antibody at room temperature. Blots were then washed 3× in TBS-T, twice in water, and then dried in the dark. Blots were imaged using a LI-COR Odyssey CLx, and images were analyzed using ImageStudioLite (LI-COR). Uncropped blots are provided in Supplementary Fig. 18.

**Rescue experiments**. Transfected HEK293T cells or primary neurons were incubated for 48 h with chemical chaperones at the indicated concentrations before analysis. Worms were kept for two–three generations on agar plates with OP50 bacteria containing either 5 mM 4-phenylbutyrate, 200 mM sorbitol, or 200 mM trehalose. Worms were then either live imaged for aggregation studies, or were subjected to heat shock paralysis or aldicarb experiments.

**S. cerevisiae experiments**. Maintenance: Wild-type strain BY4741 was maintained in YPD media (20 g/l peptone, 10 g/l yeast extract, and 20 g/l glucose). Transformed yeast strains were propagated in ura- media (6.7 g/l nitrogen base, 1.92 g/l drop out media ura-), containing either 2% glucose, 2% raffinose, or 2% galactose, or grown on ura- agar plates (6.7 g/l nitrogen base, 1.92 g/l drop out media ura-, 20 g/l agar ura-), containing either 2% glucose, or 2% galactose. Transformation: 500 μl of strain BY4741 (MATa his3Δ1 leu2Δ0 met15Δ0 ura3Δ0) grown in YPD media was transformed with 0.2 μg of empty pYES2 NT/A plasmid (control) or plasmids expressing wild-type or mutant Munc18-1b, using EZ yeast transformation according to the manufacturer's protocol (Zymo Research), and were plated on ura- agar plates containing glucose and grown over night at 30 °C. Protein expression: 5 ml of transformed yeast were grown over night in ura- medium containing 2% raffinose. The next day, 22.5 ml ura- medium containing 2% raffinose was inoculated with 1 ml of overnight culture, and cultures were grown for 4 h at room temperature. Protein production was induced by addition of 2.5 ml 20% galactose. Yeast survival was analyzed by measuring the optical density of the culture in a plate reader (Synergy H1 Hybrid Reader, BioTek), protein expression levels were analyzed by measuring GFP fluorescence in a plate reader (Synergy H1 Hybrid Reader, BioTek; excitation 488 nm, emission 510 nm), and protein aggregation was assessed by pelleting 1 ml of culture and imaging yeast in the remaining volume on a cover glass on an Eclipse 80i upright fluorescence microscope (Nikon). Toxicity assay: Overnight cultures were diluted serially (1:10)

in water, and 2 μl of culture were dotted on agar plates containing either glucose (control) or galactose (for induction of protein expression). Plates were incubated for 3 days at 20, 30, or 37 °C and were then imaged on a GelDoc XR + (BioRad).

**Antibodies**. Monoclonal antibodies used were: β-actin (A1978, Sigma), γ-adaptin (610385, BD Biosciences), GAPDH (DSHB-hGAPDH-2G7, DSHB), GFP (632381, Takara Bio Clontech), GM130 (610823, BD Biosciences), HA (901503, Biolegend), KDEL (SPA-827F, Enzo Life Sciences), Munc18-1 (610337, BD Biosciences), myc (9E10, DSHB), PSD-93 (AB_2277296, Neuromab), synaptotagmin-1 (105221, SySy), α-tubulin (12G10, DSHB). Polyclonal antibodies used were: Munc18-1 (gift from Dr. Thomas Südhof), myc (C3956, Sigma), and synaptophysin-1 (gift from Dr. Thomas Südhof).

**Quantification and statistical analysis**. Sample sizes were chosen based on preliminary experiments or similar studies performed in the past. For quantification of immunoblots, a minimum of three independent experiments were performed. For quantification of immunofluorescence microscopy images, 4000–5000 cells were counted to determine neuronal survival, and 5–10 neurons were analyzed regarding pixel intensity for the antibody uptake assay. To ensure reliable quantification across samples and images, images were recorded under the same microscope settings (objective lens and illumination intensity). Merged images were created using Photoshop (Adobe), and were analyzed using ImageJ (NIH) or Image Studio (LI-COR). For quantification of C. elegans behavior, 10–25 animals were tested for each experiment, and at least 5 independent experiments were performed. For quantification of S. cerevisiae experiments, at least three independent experiments were performed. No samples or animals were excluded from the analysis, and quantifications were performed blindly. All data are presented as the mean ± SEM, and represent a minimum of three independent experiments. Statistical parameters, including statistical analysis, significance, and n value are reported in each figure legend. Statistical analyses were performed using Prism 7 Software (GraphPad). For statistical comparison of two groups, either two-tailed Student's t test or two-way ANOVA followed by Bonferroni post hoc test was performed, as indicated in the figure legends. Based on previous studies, a normal distribution of data was assumed, with similar variance between groups that were compared. A value of $p < 0.05$ was considered statistically significant.

## Data availability

The authors declare that the data supporting the findings of this study are available within the paper and its supplementary information files.

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

## Acknowledgements

We thank Dr. Manu Sharma, Dr. Tim Ryan, and Dr. David Eliezer for helpful discussions; Dr. Thomas C. Südhof for providing antibodies; Dr. Matthijs Verhage for

providing the conditional Munc18-1 knockout mice; Dr. Cornelia Kurischko for providing yeast strains and advice; and Olga Ivleva and Dr. Tatiana Cerveira dos Santos for initial technical help. This work was supported by T32GM007739 (Weill Cornell/Rockefeller/Sloan Kettering Tri-Institutional MD PhD Program for D.A.), R01-GM095674 (J.S.D.), R01-NS102181 (J.B.), the Epilepsy Foundation & American Epilepsy Society (J.B.), the Leon Levy Foundation (J.B.), and the Sanofi Innovation Awards Program (J.B.).

## Author contributions

N.G.L.G. and A.P. performed most of the experimental work and analyzed data. D.A. assisted with Dendra2 photoconversion experiments, P.K. with yeast studies, K.E.C. with in vitro experiments, and R.W. with *C. elegans* studies. J.B. coordinated the study and designed the experiments. J.S.D. and J.B. wrote the manuscript. All of the authors approved the manuscript.

## Additional information

**Competing interests:** The authors declare no competing interests.

