## [Peer Review File · Nature Communications]

Reviewers' comments:

Reviewer #1 (Remarks to the Author):

In this paper, the authors explore the effects of disease-causing mutations in the human Munc-18 protein, which regulates synaptic release. Disease is caused by mutations in a single allele, and had been thought to arise by haploinsufficiency. The authors here make the argument that these disease mutations function in a dominant negative fashion. By examining functional neurotransmission, protein aggregation, and effects of chaperons in cultured neurons and other mammalian cell types, in *C. elegans*, and in yeast, the authors provide reasonably compelling evidence for their hypothesis. The results, therefore, suggest an alternative hypothesis for disease etiology, which may point the way to clinical therapeutics.

Diseases of the nervous system are notoriously difficult to treat, and this is often the case because a lost function needs to be restored- a difficult feat. However, interfering mutations provide perhaps for more technically approachable strategies, based on blocking dominant-negative interactions. This paper provides a proof of principle for this idea in the context of Munc-18 mutations that lead to human disease. The paper is clearly written, the results are statistically robust, and the interpretations make sense. Particularly striking is the ability of small molecule chaperones to reverse deficits in the cell culture and *C. elegans* in vivo models.

I do not have any major concerns, although I do feel that the authors should at least discuss more, and perhaps perform additional experiments, addressing the issue that nearly all their studies are in an overexpression context. In *C. elegans*, for example, CRISPR technology is already available, and introducing single copy heterozygous modifications should be feasible, and would likely reflect the disease paradigm more accurately. The same holds for some of the cell culture settings, and certainly in yeast. I don't think they need to repeat all their studies, but even one or two example of this approach would go a long way in making the data that more relevant to disease.

Reviewer #2 (Remarks to the Author):

Guilberson et al. undertake a variety of functional and biochemical studies to try to understand why mutations in Munc18-1 lead to neuronal dysfunction. In general the work supports the contention that mutant Munc18-1 is leads to alterations in synaptic communication, an unstable protein that is not degraded normally (and aggregates) and improving solubility of the proteins restores protein function. Several statements regarding data interpretation are imprecise and missing analysis of specific mutants (that could support or refute the underlying contentions made the authors) need rectification.

The authors begin by studying a worm with mutation in the putative ortholog and find that the animals are very uncoordinated. Restoring WT Munc18-1 rescues this phenotype. It

appears that the human Munc18-1 is employed here although this information could be more obviously provided. Transgenic expression of point mutant Munc18-1 also rescue locomotor behavior (figure 1b) albeit less well than the WT protein. So the mutant proteins clearly have retained some WT activity. In the aldicarb assay some transgenic mutant lines behave like WT rescues (e.g., P479L) while others do not. Paralysis in response to heat similarly varies between different mutant lines. Ditto uptake of synaptotagmin-1 antibody upon high K⁺ stimulation.

The authors need to show where endogenous Munc18-1 is normally expressed in the worm. If it is expressed in all neurons (both GABAergic and cholinergic) then the interpretation of the aldicarb work requires revision.

The uptake studies suggest that Munc18-1 impairs recycling of synaptic vesicles. Since neurotransmitter release is not being studied by this assay, the statement that "... Munc18-1 impairs neurotransmitter release..." is unsubstantiated. The authors should use an electrophysiological assay in the primary neuron cultures to directly probe this issue.

The authors use lentivirus to overexpress WT and mutant Munc18-1 in Munc18-1 KO neurons and find exceptionally low expression levels for most but not all of the constructs. The authors do not look at the P479L mutant (which when expressed in worms completely rescues the aldicarb phenotype) and it should be. In addition an assessment of message expression levels is encouraged since low levels of the transgene might be occurring at the transcription level.

The stability assays involve incubation with CHX for 3, 8 and 24 hours. While group differences are found, this assay is problematic because neurons become quite unhappy when protein synthesis is blocked for long periods. If cell death is evoked by 8 and 24 hours treatment with CHX, the interpretation of these observations becomes problematic. Better approaches are radioactive amino acid pulse-chase followed by immunoprecipitation of the protein or an optical techniques such as dendra2 photoconvertible fusion protein. These presented data are not strong, notwithstanding the demonstration that the protein aggregates.

The authors provide evidence that mutant Munc18-1 binds the WT proteins, leading to its aggregation and loss of function. The effect sizes are subtle. In an interesting test of this notion the authors express mutant proteins in N2 worms and see a dominant effect. The P479L mutant is curious. On a Munc18-1 null background, transgenic expression of the P479L mutant has normal aldicarb response (Figure 1d). On the WT background the P479L mutant an abnormal response. The reconciliation of these disparate observations is not only unclear; it undermines the hypothesis the authors are proposing. It would also be informative to study the P335L mutant since this protein does not aggregate (figure 3 panel I). If it too shows a "dominant negative" effect the authors have some explaining to do.

The authors provide evidence that chemical chaperones improve the solubility and functionality of mutant Munc18-1. The effects are fairly subtle (particularly at the biochemical level).

Authors' response to reviewers' comments for Guiberson et al., "Mechanism-based rescue of Munc18-1 dysfunction in varied encephalopathies by chemical chaperones *in vitro* and *in vivo*" and changes instituted into the manuscript as a result

We thank the reviewers for their insightful comments. As described below, we have added a number of additional experiments, in particular to expand and strengthen the understanding of the disease mechanism underlying Munc18-1 mutations as well as the presented rescue strategies. Moreover, we made significant changes in the text in order to respond to the reviewers' and editors' comments.

Briefly, the following new data were added to the revised manuscript:

- 1) Generation of new CRISPR/Cas9-edited knock-in worms and then testing our previous studies in transgenic worms in these new worm disease models (Fig. 1g-i, Fig. 7d and 7e, Supplementary Fig. 2e-g, and Supplementary Fig. 17b and 17c).
- 2) Basic characterization of neuronal activity in primary mouse neurons expressing Munc18-1 variants using a multi-electrode array (Fig. 2a-g and Supplementary Fig. 3c-e).
- 3) Further demonstration that our data in primary neurons are achieved under endogenous-like Munc18-1 protein levels, and not under overexpression (Supplementary Fig. 3a and 3b).
- 4) Demonstration that alterations in transcript levels are not responsible for the observed reduction in mutant Munc18-1 protein levels (Supplementary Fig. S4a-c).
- 5) Analysis of accelerated mutant Munc18-1 turnover using a Dendra2 photoconversion assays (Fig. 3c-f and Supplementary Fig. 4g-j) to complement the previously shown cycloheximide chase assays.

We hope that the reviewers and editors will find these substantial changes convincing, and that the paper can be deemed acceptable for publication.

Reviewers' comments:

Reviewer #1 (Remarks to the Author):

In this paper, the authors explore the effects of disease-causing mutations in the human Munc-18 protein, which regulates synaptic release. Disease is caused by mutations in a single allele, and had been thought to arise by haploinsufficiency. The authors here make the argument that these disease mutations function in a dominant negative fashion. By examining functional neurotransmission, protein aggregation, and effects of chaperons in cultured neurons and other mammalian cell types, in *C. elegans*, and in yeast, the authors provide reasonably compelling evidence for their hypothesis. The results, therefore, suggest an alternative hypothesis for disease etiology, which may point the way to clinical therapeutics.

Diseases of the nervous system are notoriously difficult to treat, and this is often the case because a lost function needs to be restored- a difficult feat. However, interfering mutations provide perhaps for more technically approachable strategies, based on blocking dominant-negative interactions. This paper provides a proof of principle for this idea in the context of

Munc-18 mutations that lead to human disease. The paper is clearly written, the results are statistically robust, and the interpretations make sense. Particularly striking is the ability of small molecule chaperones to reverse deficits in the cell culture and *C. elegans* in vivo models.

We thank the reviewer for the very positive assessment of our work, and provide below a point-by-point response in blue. Changes made in the manuscript are highlighted in blue font.

I do not have any major concerns, although I do feel that the authors should at least discuss more, and perhaps perform additional experiments, addressing the issue that nearly all their studies are in an overexpression context. In *C. elegans*, for example, CRISPR technology is already available, and introducing single copy heterozygous modifications should be feasible, and would likely reflect the disease paradigm more accurately. The same holds for some of the cell culture settings, and certainly in yeast. I don't think they need to repeat all their studies, but even one or two example of this approach would go a long way in making the data that more relevant to disease.

We fully agree with the reviewer that overexpression may introduce artefacts. We had therefore included quantitation of lentivirally expressed protein levels of the myc-tagged Munc18-1 in primary cortical mouse neurons in our previous manuscript (Supplementary Fig. 3a). We have now amended this analysis using a titration study (new Supplementary Fig. 3b), highlighting again the lack of overexpression of Munc18-1 in primary neurons. In fact, while we routinely use 20 μ l of concentrated lentivirus for infections, even doubling this amount does not result in overexpression of Munc18-1 (Supplementary Fig. 3b), suggesting a saturation effect.

In addition, we agree that introduction of single copy modifications in *C. elegans* likely reflects the disease paradigm more accurately. We have therefore generated CRISPR/Cas9-edited *C. elegans*, carrying the mutations P334L and R405H in the *unc-18* gene (new Supplementary Fig. 2e and 2f). Generation of P479L, G544D and G544V mutants was also attempted, but screening 300-400 worms for each of these mutations for a positive integration event was unsuccessful. Yet, we feel that the P334L and R405H knock-in worms represent the spectrum of phenotypes in our study very well, with P334L having a rather mild and R405H having a severe phenotype (see below).

Using our CRISPR-edited *unc-18* knock-in strains, we have repeated assays for locomotion, neurotransmitter release, and heat shock paralysis. Similar to our transgenic worm models, P334L and R405H worms showed reduced locomotion and an exaggerated heat shock paralysis compared to N2 worms (Fig. 1g and 1i). Noteworthy, the mutant phenotype of our knock-in worms was slightly stronger compared to the transgenic worms, suggesting that overexpression of mutant UNC-18 variants partially compensated for the loss of function (compare panels b, d, and e with panels g and i in Fig. 1). The aldicarb assay was interesting, in that R405H showed the expected deficits, but P334L gave rise to a hypersensitive phenotype, suggesting an increase in neurotransmitter release. In fact, the P335 residue is unique in that it is located within domain 3a of Munc18-1, a region that undergoes conformational changes to mediate binding to syntaxin-1, VAMP2, or the SNARE-complex, and P335 has been proposed to serve as hinge point of a bent hairpin in Munc18-1, allowing the helical hairpin to access both extended and bent

conformations and thereby regulating the binding to its effector protein syntaxin-1, the SNARE-complex, and neurotransmitter release. In line with our data showing an increase in neurotransmitter release, P335A has been reported to result in an extended alpha-helical conformation and accelerates lipid mixing (Parisotto et al, J Biol Chem, 2014). With this requirement for flexibility surrounding residue P335, mutations in this domain are expected to be less deleterious compared to mutations in other domains of Munc18-1, and this is likely the reason for the intermediate-level deficits in synaptic vesicle cycling, protein stability and turnover, and aggregation that we have observed (Figs. 1-7). We have now added this observation to the discussion.

We have also analyzed the effect of chemical chaperones on these new knock-in worms and found a similar rescue effect compared to the transgenic *C. elegans* (Fig. 7d and 7e, and Supplementary Fig. 17b and 17c).

Reviewer #2 (Remarks to the Author):

Guilberson et al. undertake a variety of functional and biochemical studies to try to understand why mutations in Munc18-1 lead to neuronal dysfunction. In general the work supports the contention that mutant Munc18-1 leads to alterations in synaptic communication, an unstable protein that is not degraded normally (and aggregates) and improving solubility of the proteins restores protein function.

We thank the reviewer for this overall positive assessment of our work, and provide below a point-by-point response in blue.

Several statements regarding data interpretation are imprecise and missing analysis of specific mutants (that could support or refute the underlying contentions made the authors) need rectification.

We thank the reviewer for pointing this out. We have clarified the imprecise data interpretation and have added the P334L mutation to the *C. elegans* experiments. All changes made to the manuscript are highlighted in blue font.

The authors begin by studying a worm with mutation in the putative ortholog and find that the animals are very uncoordinated. Restoring WT Munc18-1 rescues this phenotype. It appears that the human Munc18-1 is employed here although this information could be more obviously provided.

We apologize for the lack of clarity. For all *C. elegans* experiments, the *C. elegans* ortholog UNC-18 was used to (1) maintain expression of a protein that is adapted to protein folding at 20°C (worm growth conditions) versus 37°C in humans or mice, and to (2) maintain endogenous

protein interactions due to slight sequence variations between the worm UNC-18 and human Munc18-1 protein sequence (see Supplementary Fig. 2a).

Transgenic expression of point mutant Munc18-1 also rescue locomotor behavior (figure 1b) albeit less well than the WT protein. So the mutant proteins clearly have retained some WT activity. In the aldicarb assay some transgenic mutant lines behave like WT rescues (e.g., P479L) while others do not. Paralysis in response to heat similarly varies between different mutant lines. Ditto uptake of synaptotagmin-1 antibody upon high K⁺ stimulation.

The reviewer is correct in pointing out that mutant UNC-18 partially rescues locomotor behavior. Based on our molecular data, this can be explained by remaining functional mutant protein that does not aggregate and is not degraded. Also, as explained below and as the reviewer correctly points out, transgenic *C. elegans* express 4-20-fold levels of endogenous UNC-18. Thus, any remaining functional proteins levels will be exaggerated under these conditions. In fact, P479L has some of the highest expression levels among all mutants analyzed (Supplementary Fig. 2c), which likely accounts for the better rescue effect of *unc-18* null worms in the aldicarb assay. We have now addressed this potential problem by generation of CRISPR-edited *C. elegans* strains (see below).

The authors need to show where endogenous Munc18-1 is normally expressed in the worm. If it is expressed in all neurons (both GABAergic and cholinergic) then the interpretation of the aldicarb work requires revision.

UNC-18 is present at all chemical synapses within *C. elegans* and is obligate for synaptic transmission regardless of neurotransmitter type (Weimer et al, Nat Neurosci, 2003). Sensitivity to cholinesterase inhibitors provides a simple proxy for acetylcholine secretion (by specifically targeting the enzyme that breaks down acetylcholine), and cholinergic neurons comprise over 1/3 of the worm nervous system (~120 neurons functioning predominantly as motor neurons). In contrast, there are only 19 GABAergic motor neurons. Thus, although *unc-18* mutants will have disruption in both acetylcholine and GABA secretion, the aldicarb assay will be highly sensitive to a decrease in acetylcholine secretion. Similar results have been observed with other ubiquitous synaptic proteins such as UNC-13, Synaptotagmin 1 and SNAREs (Miller et al, PNAS, 1996; Yook et al, Genetics, 2001).

The uptake studies suggest that Munc18-1 impairs recycling of synaptic vesicles. Since neurotransmitter release is not being studied by this assay, the statement that "... Munc18-1 impairs neurotransmitter release..." is unsubstantiated. The authors should use an electrophysiological assay in the primary neuron cultures to directly probe this issue.

We agree and have now included measurements of neuronal activity using a multielectrode array, including mean firing rate, network burst frequency and duration (new Fig. 2a-g and Supplementary Fig. 3c-e). Similar to the antibody uptake assay, we find severe deficits for R406H, P480L, G544D, and G544V. Compared to WT Munc18-1, P335L caused no change or a

non-significant increase in these parameters, in agreement with published studies (Parisotto et al, J Biol Chem, 2014). We have added a discussion of these findings to the discussion section.

The authors use lentivirus to overexpress WT and mutant Munc18-1 in Munc18-1 KO neurons and find exceptionally low expression levels for most but not all of the constructs. The authors do not look at the P479L mutant (which when expressed in worms completely rescues the aldicarb phenotype) and it should be.

We apologize for the lack of clarification on the mutants. P479L in the worm UNC-18 protein is homologous to P480L in the human Munc18-1 sequence (see also Supplementary Fig. 2a and 2b). Our measurements in primary mouse neurons clearly indicate increased turnover and aggregation of P480L compared to WT Munc18-1, accompanied by defects in synaptic function (Fig. 3 and 4). As outlined above, the lack of effect of worm P479L in transgenic *C. elegans* in the aldicarb assay is likely due to overexpression of the mutant protein, resulting in elevated levels of the functional protein fraction that compensates for the lack of endogenous UNC-18, or due to the fact that the aldicarb assay may not be sensitive enough to detect the synaptic effects of the P479L mutation. Yet, P479L-expressing worms revealed significant impairments in worm locomotion and an exaggerated heat shock paralysis (Fig. 1).

In addition an assessment of message expression levels is encouraged since low levels of the transgene might be occurring at the transcription level.

We thank the reviewer for this suggestion. We have now performed qPCR analysis of transcript levels in our primary neurons, and find no change in mRNA levels between WT and mutant Munc18-1 variants (new Supplementary Fig. S4a-c).

The stability assays involve incubation with CHX for 3, 8 and 24 hours. While group differences are found, this assay is problematic because neurons become quite unhappy when protein synthesis is blocked for long periods. If cell death is evoked by 8 and 24 hours treatment with CHX, the interpretation of these observations becomes problematic. Better approaches are radioactive amino acid pulse-chase followed by immunoprecipitation of the protein or an optical techniques such as dendra2 photoconvertible fusion protein. These presented data are not strong, notwithstanding the demonstration that the protein aggregates.

We thank the reviewer for this valuable suggestion. While CHX for 8 and 24 hours has been shown to be non-toxic to neurons (Sheehan et al., J Neurosci, 2016), we agree that CHX may block translation of proteins that interact with Munc18-1 and therefore affect its turnover. We thus followed the reviewer's suggestion and performed Dendra2 photoconversion experiments in primary neurons and in transfected HEK293T cells expressing Munc18-1:Dendra2 fusion proteins (new Fig. 3c-f and Supplementary Fig. 4g-j). We found a similar significant increase in turnover for mutant Munc18-1 compared to WT Munc18-1 in both systems.

The authors provide evidence that mutant Munc18-1 binds the WT proteins, leading to its aggregation and loss of function. The effect sizes are subtle.

Our analysis of total protein levels and Triton X-100 soluble Munc18-1 levels in primary neurons indicate that functional Munc18-1 levels are in the range of 30-40% in a heterozygous condition (Fig. 5a-d). Larger effects on Munc18-1 levels are predicted to be non-viable, as evidenced by embryonic lethality in Munc18-1 knockout mice and flies.

In contrast to mice and flies, worms lacking UNC-18 are viable albeit severely paralyzed. As worms do not need a functional nervous system to survive and reproduce, this model system has been useful for studying critical synaptic proteins such as UNC-13, UNC-18, and synaptotagmin, for which mammalian null mutants are not viable.

In an interesting test of this notion the authors express mutant proteins in N2 worms and see a dominant effect. The P479L mutant is curious. On a Munc18-1 null background, transgenic expression of the P479L mutant has normal aldicarb response (Figure 1d). On the WT background the P479L mutant an abnormal response. The reconciliation of these disparate observations is not only unclear; it undermines the hypothesis the authors are proposing.

We apologize for not making this point more clear. Significantly more UNC-18 P479L protein is expressed on the *unc-18* null background compared to the wild-type N2 background (compare Supplementary Figs. 2c and 11a). As described above, high amount of overexpression is able to at least partially compensate for the mutant phenotype and results in rescue due to remaining high functional mutant protein amounts. This is in line with our observation that CRISPR/Cas9-generated knock-in worms show more severe phenotypes on the *unc-18* null background (Fig. 1). In addition, we have demonstrated robust and significant impairments of P479L in the locomotion assay and heat shock paralysis experiments in all worm strains analyzed (Fig. 1b, 1e, 1g, 1i, 5e-g) and in primary cortical mouse neurons (Figs. 2-5 and the accompanying Supplementary figures).

It would also be informative to study the P335L mutant since this protein does not aggregate (figure 3 panel I). If it too shows a “dominant negative” effect the authors have some explaining to do.

We have followed the reviewer’s suggestion and have generated CRISPR-edited P334L knock-in worms, equivalent to the P335L mutation in humans (Supplementary Fig. 2b and 2e). These worms show dysfunction in heat shock response and locomotion (Fig. 1g and 1i), and an increase in neurotransmitter release in the aldicarb assay (Fig. 1h), similar to our primary neuron data (Fig. 2d-g).

Overall, as the reviewer points out, the P335L mutation shows a milder phenotype compared to the other mutations, including no or a lower amount of aggregation in primary neurons (Fig. 3c), in yeast (Fig. 3i), and in transfected Neuro2a cells (Supplementary Fig. 6). We have no

indication of a dominant negative effect of P335L on WT Munc18-1 regarding total Munc18-1 levels (Fig. 5a and 5b), solubility (assessed by limited proteolysis in Supplementary Fig. 10b-e, and by Triton X-100 solubility in Fig. 5d and Supplementary Fig. 10a) or aggregation (Fig. 3c, 4c, 4h, and Supplementary Fig. 6). The P335 residue is interesting and unique. P335 is located within domain 3a of Munc18-1 (Supplementary Fig. 1b), a region that undergoes conformational changes to mediate binding to syntaxin-1, VAMP2, or the assembled SNARE-complex. P335 has been proposed to serve as a hinge point of a bent hairpin in Munc18-1, allowing the helical hairpin to access both extended and bent conformations and thereby regulating the binding to its effector protein syntaxin-1, the SNARE-complex, and neurotransmitter release. In line with our data showing an increase in neurotransmitter release (Fig. 1h and 2d-g), P335A has been reported to result in an extended alpha-helical conformation and accelerates lipid mixing (Parisotto et al, J Biol Chem, 2014). With this requirement for flexibility surrounding P335, mutations in this domain are expected to be less deleterious compared to mutations in other domains of Munc18-1, and this is likely the reason for the intermediate-level deficits in synaptic vesicle cycling, protein stability and turnover, and aggregation that we have observed, and the lack of a dominant-negative effect as demonstrated for the other missense mutations. We thus classify this unique mutation as a loss-of-function mutation regarding the loss of its ability to undergo conformational changes in domain 3a, and have added a paragraph in the discussion to highlight this.

The authors provide evidence that chemical chaperones improve the solubility and functionality of mutant Munc18-1. The effects are fairly subtle (particularly at the biochemical level).

We would like to point out that our analysis of total protein levels and Triton X-100 soluble Munc18-1 levels in primary neurons demonstrate that total functional Munc18-1 levels in a heterozygous condition are in the range of 30-40% (Fig. 5a-d). Thus, stabilizing Munc18-1 levels by 10-20% is expected to be sufficient to push functional protein levels over the critical 50% threshold. The three chemical chaperones tested here provide a 10-60% increase in total protein levels in primary neurons (Fig. 6a) in addition to a 20-100% increase in Triton X-100 solubility in primary neurons (Fig. 6b), accomplishing this goal. This is furthermore evidenced by rescue of neuron function in primary neurons and in worms (Fig. 6c, 6d, and Fig. 7a-f), and by rescue of the subcellular localization of UNC-18 G544D in *C. elegans* (Figure 7g).

REVIEWERS' COMMENTS:

Reviewer #1 (Remarks to the Author):

The authors have done an outstanding job in responding to reviewer concerns, and I think that the paper is on very solid scientific footing.

Reviewer #2 (Remarks to the Author):

The authors have done an excellent job addressing all my concerns and I recommend publication

REVIEWERS' COMMENTS:

Reviewer #1 (Remarks to the Author):

The authors have done an outstanding job in responding to reviewer concerns, and I think that the paper is on very solid scientific footing.

Reviewer #2 (Remarks to the Author):

The authors have done an excellent job addressing all my concerns and I recommend publication

We thank both reviewers for their enthusiastic comments.